# Intriguing Properties of Large Language and Vision Models

## Abstract

Recently, large language and vision models (LLVMs) have received significant attention and development efforts due to their remarkable generalization performance across a wide range of tasks requiring perception and cognitive abilities. A key factor behind their success is their simple architecture, which consists of a vision encoder, a projector, and a large language model (LLM). Despite their achievements in advanced reasoning tasks, their performance on fundamental perception-related tasks (e.g., MMVP) remains surprisingly low. This discrepancy raises the question of how LLVMs truly perceive images and exploit the advantages of the vision encoder. To address this, we systematically investigate this question regarding several aspects: *permutation invariance*, *robustness*, *math reasoning*, *alignment preserving* and *importance*, by evaluating the most common LLVM's families (i.e., LLaVA) across 10 evaluation benchmarks. Our extensive experiments reveal several intriguing properties of current LLVMs: (1) they internally process the image in a global manner, even when the order of visual patch sequences is randomly permuted; (2) they are sometimes able to solve math problems without fully perceiving detailed numerical information; (3) the cross-modal alignment is overfitted to complex reasoning tasks, thereby, causing them to lose some of the original perceptual capabilities of their vision encoder; (4) the representation space in the lower layers ($< 25\%$) plays a crucial role in determining performance and enhancing visual understanding. Lastly, based on the above observations, we suggest potential future directions for building better LLVMs and constructing more challenging evaluation benchmarks.

## 1 Introduction

Large Language and Vision Models (LLVMs)(Liu et al., 2024c;b; Lee et al., 2024c;b) have demonstrated remarkable generalization capabilities across a wide variety of tasks, including coding and mathematics, showcasing their potential for practical applications. These impressive advancements have been achieved through a straightforward yet effective architecture based on the concept of *model-stitching*(Lenc & Vedaldi, 2015; Bansal et al., 2021). This approach integrates a pre-trained vision encoder (Radford et al., 2021) with a pre-trained large language model (LLM) (Touvron et al., 2023; Zheng et al., 2023b) via a simple cross-modal alignment module. This method significantly benefits from the power of well-established pre-trained representations. Consequently, this structure has become the *de facto* standard in the field, extending into other modality domains such as video, audio, and unified modalities (Xie et al., 2024; Erfei Cui, 2024).

Despite their significant generalization performance, recent studies have revealed several interesting phenomena regarding LLVMs. For instance, they struggle with tasks that are easy for humans to perceive (e.g., MMVP (Tong et al., 2024), BLINK (Fu et al., 2024)) and have limited understanding of domain-specific images (Zhai et al., 2024; Verma et al., 2024). In contrast to the computer vision domain, where demystifying the properties of vision encoders has been more thoroughly explored (Naseer et al., 2021; Kim et al., 2023; Vishniakov et al., 2023), the underlying properties of LLVMs are still largely under-explored. Therefore, in this work, we scrutinize the current *de facto* structure of LLVMs under various partial conditions, such as *permutation*, *occlusion*, and *synthetic images*.

In this paper, we systematically conduct a comprehensive evaluation of widely used LLVM families, specifically the `LLaVA`-series, across 10 diverse benchmarks, which encompass tasks such as math, chart, and basic perception. Our extensive experiments reveal several intriguing properties of current LLVMs, which we summarize as follows:

- In LLVMs, the visual patch tokens processed through the projector exhibit varying magnitudes of localized visual information. Remarkably, even when the order of these patch sequences is randomly shuffled before being fed into the LLM, the performance does not significantly degrade. For instance, in the case of `LLaVA 1.5` (Li et al., 2024c), the average performance drop across 10 benchmarks is 0.19 ($< 1\%$), indicating that LLVMs exhibit permutation-invariant properties.

- LLVMs effectively handle tasks when given synthetic versions of the MathVista (Lu et al., 2023) dataset, with only a small performance decline (1.8% for `LLaVA 1.5`). Furthermore, we discovered that, in certain scenarios, LLVMs can solve problems even without access to the full image, including detailed numerical and chart elements.

- Following alignment and visual instruction tuning, LLVMs fail to preserve their initial perceptual capacities, with up to a 20% drop in image classification tasks (e.g., CIFAR-100 (Krizhevsky et al., 2009)), a phenomenon known as catastrophic forgetting (Zhai et al., 2024). Furthermore, they struggle to understand shared-world concepts within the representation space, according to the platonic representation hypothesis (Huh et al., 2024).

- Our analysis of model behavior reveals that LLVMs tend to concentrate more on the central region of the image. Furthermore, the lower layers in LLVM architectures are crucial for better generalization. In these layers (i.e., the bottom 20% of the LLM layers), the model primarily processes visual information, while the higher layers focus on interpreting the text.

In addition to our findings, we present and discuss several points regarding LLMs and evaluation benchmarks. Specifically, we highlight the need to develop more interactive and complex evaluation benchmarks to mitigate selection bias Zheng et al. (2023a) and improve applicability to real-world scenarios. Furthermore, when developing new LLMs, it is crucial to preserve cross-modal alignment. We hope that our findings will assist other ML researchers and engineers in building a new paradigm for LLMs.

## 2 RELATED WORKS

**Large Language and Vision Models.** Recent advancements in LLVMs have predominantly adopted simplistic yet highly effective architectures, notably through the model-stitching concept. Numerous prior studies have introduced various design modifications to bridge the performance gap with closed-source LLVMs (OpenAI, 2023; Anthropic, 2024). These efforts include focusing intently on high-resolution processing (Li et al., 2024e; Liu et al., 2024b; Shi et al., 2024), implementing locality-enhanced projectors (Cha et al., 2024), and incorporating knowledge embeddings (Lee et al., 2024c), layer traversal technique (Lee et al., 2024b) and leveraging a diverse array of vision encoders (Lu et al., 2024; Tong et al., 2024) have also been explored. Additionally, integrating external, task-specific computer vision modules (Lee et al., 2024d;e; Jiao et al., 2024; Lai et al., 2024) and incorporating different modalities — including video and audio (Wang et al., 2024; Li et al., 2024b; Erfei Cui, 2024; Xie et al., 2024) — have expanded the models' capabilities. Moreover, enabling the handling of interleaved input formats (Li et al., 2024c; Xue et al., 2024) has further broadened the versatility of these models. While these models have been developed based on a simplistic structure, *model-stitching*, the effectiveness of this architecture remains under-explored.

**Investigating Intriguing Properties of LLVMs.** Alongside these advancements, recent studies have investigated and uncovered several crucial properties of current LLVMs. For instance, some studies have rigorously evaluated LLVMs on basic perception tasks that are trivially easy for humans by introducing "blind" pairs of image datasets (Tong et al., 2024; Fu et al., 2024; Rahmanzadehgervi et al., 2024). Other studies have explored cross-modal alignment by focusing on domain-specific visual capabilities (Verma et al., 2024) and examining the alignment of representation spaces across modalities between independently pre-trained LLMs and vision encoders (Li et al., 2024d; Huh

et al., 2024). Zhai et al. (2024) examine the phenomenon of catastrophic forgetting in LLVMs within the context of image classification tasks. Additional studies (Zhou et al., 2023b; Chen et al., 2024b) analyze the persistent issue of object hallucination in LLVMs. Moreover, research has explored spatial reasoning capabilities (Kamath et al., 2023). While vision encoders (e.g., ViT (Dosovitskiy, 2020), DeiT (Touvron et al., 2021)) in the computer vision field have been rigorously examined across a wide range of image settings, the study of these intriguing properties in LLVMs remains relatively under-explored. In this paper, we aim to address this by conducting an in-depth investigation into LLVMs, examining their permutation invariance, robustness, alignment preservation, and importance in scenarios involving occluded and synthesized images.

## 3 DEMYSTIFYING INTRIGUING PROPERTIES OF LVLMS

In this section, we explore the intriguing properties of current LLVMs that have *de facto* structure of *modal-stitching* in terms of various aspects: permutation invariance, robustness to occlusion, synthetic data, alignment preserving, and importance.

### 3.1 BACKGROUND

**Overview of LVLM.** Current LVLMs $\mathcal{M}$ have widely adopted the *model-stitching* architecture, which consists of three main components: a pre-trained vision encoder $f_\text{V}$, a projector $f_\text{P}$, and a pre-trained LLM $f_\text{L}$. The overall model is represented as $\mathcal{M} = f_\text{L} \circ f_\text{P} \circ f_\text{V}$. The vision encoder $f_\text{V}$ converts the input image $I \in \mathbb{R}^{3 \times H \times W}$ into visual features $\mathcal{F}_v \in \mathbb{R}^{N \times d_v}$, where $N = HW/P^2$ is the number of visual features, $P$ is the patch size, and $d_v$ is the dimension of the vision encoder's output. The projector $f_\text{P}$ transforms these visual features $\mathcal{F}_v$ into visual patch tokens $\mathbf{X}_\text{V} \in \mathbb{R}^{N \times d_l}$ in the representation space of the LLM, where $d_l$ is the embedding dimension of the LLM. This mapping allows the LLM to perceive and conceptually understand the given image. The LLM $f_\text{L}$ produces an appropriate response $\mathbf{Y} = \{y_i\}_{i=1}^{L_\mathbf{Y}}$ in an autoregressive manner, given both the visual patch tokens $\mathbf{X}_\text{V}$ and the text tokens $\mathbf{X}_\text{T} \in \mathbb{R}^{L_\mathbf{T} \times d_l}$, where $L_\mathbf{T}$ denotes the length of the input text sequence, and $L_\mathbf{Y}$ is the length of the output sequence. The probability of generating the response is given by:

$$p(\mathbf{Y} \mid \mathbf{X}_\text{V}, \mathbf{X}_\text{T}) = \prod_{i=1}^{L_\mathbf{Y}} p(y_i \mid \mathbf{X}_\text{V}, \mathbf{X}_\text{T}, y_{<i}) \tag{1}$$

### 3.2 EVALUATION SETUP

**Evaluation Benchmarks.** To ensure a comprehensive and rigorous evaluation, we employ 10 standard and widely adopted benchmarks: MMVP (Tong et al., 2024), Q-Bench (Wu et al., 2023), MME (Fu et al., 2023), MMStar (Chen et al., 2024a), MM-Vet (Yu et al., 2023), LLaVA-W (Liu et al., 2024c), MathVista (Lu et al., 2023), SQA-IMG (Lu et al., 2022a), ChartQA (Masry et al., 2022), and AI2D (Kembhavi et al., 2016). Detailed descriptions of each dataset are provided in Appendix J.

**Evaluation Models.** Recently, a large number of LVLM models have been actively introduced, owing to their remarkable flexibility and versatility across multiple domains. Consequently, it is challenging and inefficient to conduct holistic evaluations on all LVLMs. Therefore, we select most standard LLVMs: `LLaVA-1.5-7B` (Li et al., 2024c), `LLaVA-NeXT-7B` (Liu et al., 2024b), and `LLaVA-OneVision-8B` (Li et al., 2024b). For our customized experiments, before evaluating LVLMs under diverse settings (e.g., occlusion), we first attempt to reproduce the baseline performance of LVLMs on 10 evaluation benchmarks. To do this, we implement our customized evaluation toolkits by referring to the code of `UniBench`[1] (Al-Tahan et al., 2024). Detailed descriptions of each model are provided in Appendix K.

---

[1] https://github.com/facebookresearch/unibench

| LLVMs | MMVP | Q-Bench | MME | MMStar | MM-Vet | LLaVA$^W$ | MathVista | SQA$^I$ | ChartQA | AI2D | Avg. $\Delta$ |
|---|---|---|---|---|---|---|---|---|---|---|---|
| LLaVA-1.5 | 34.67 | 59.73 | 1850.07 | 34.20 | 31.50 | 67.50 | 24.70 | 65.59 | 16.92 | 53.34 | |
| + Perm. | 36.00 | 59.60 | 1874.60 | 33.33 | 30.40 | 66.20 | 21.20 | 65.44 | 14.08 | 52.69 | ▼ 0.59 |
| | (▲ 1.33) | (▼ 0.13) | (▲ 24.53) | (▼ 0.87) | (▼ 1.10) | (▼ 1.30) | (▼ 3.50) | (▼ 0.15) | (▼ 2.84) | (▼ 0.65) | |
| LLaVA-NeXT | 36.67 | 63.55 | 1874.42 | 37.80 | 43.50 | 75.50 | 32.00 | 62.12 | 66.06 | 64.02 | |
| + Perm. | 37.33 | 62.54 | 1890.19 | 36.87 | 43.40 | 75.80 | 21.70 | 62.12 | 34.55 | 64.02 | ▼ 2.71 |
| | (▲ 0.67) | (▼ 1.00) | (▲ 15.78) | (▼ 0.93) | (▼ 0.10) | (▲ 0.30) | (▼ 10.30) | (▼ 0.00) | (▼ 31.51) | (▼ 0.00) | |
| LLaVA-OneVision | 60.67 | 77.26 | 1982.5 | 59.87 | 57.80 | 87.40 | 61.80 | 94.00 | 93.52 | 81.25 | |
| + Perm. | 59.33 | 76.99 | 1964.3 | 54.93 | 47.60 | 82.30 | 53.50 | 89.24 | 58.26 | 75.58 | ▼ 9.40 |
| | (▼ 1.33) | (▼ 0.27) | (▼ 18.2) | (▼ 4.93) | (▼ 10.20) | (▼ 5.10) | (▼ 8.30) | (▼ 4.76) | (▼ 35.26) | (▼ 5.67) | |

Table 1: Results of drop ratio ($\Delta$) when random permutation is applied. We run five experiments.

## 3.3 Do LLVMs Perceive Images Globally?

Current LVLMs commonly adopt ViT (Dosovitskiy, 2020)-based vision encoders, such as CLIP ViT (Radford et al., 2021) and SigLIP (Zhai et al., 2023), making their image perception dependent on these encoders. Specifically, ViT is designed to learn interactions across all image patches, providing properties (Naseer et al., 2021; Vishniakov et al., 2023) such as *permutation invariance* and *robustness to occlusion*. This raises the question of whether these ViT properties might transfer to current LVLM models.

**Each visual patch token encapsulates localized visual information.** We first investigate whether each visual patch token $\mathbf{X}_V$ from the projector $f_P$ captures a localized understanding of the patch area corresponding to its position in the image. Specifically, given an image $I$, the projector outputs $N$ visual patch tokens (e.g., $N = 576$ for LLaVA-1.5-7B). We then select a single token (removing all others) and feed it into the LLM $f_L$. To quantify this, we define the patch information loss (PIL) as the ratio of the performance drop to the original performance. However, performing computations on each individual visual token is computationally intensive, especially for models such as LLaVA 1.5-7B that process 576 visual tokens arranged in a $24 \times 24$ grid of patches. To accelerate computation and reduce complexity, we aggregate the original

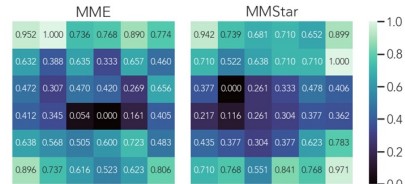

Figure 1: We demonstrate the extent to which group-wise visual tokens capture region-specific information (PIL) for LLaVA-1.5-7B on the MMStar (Chen et al., 2024a) and MME (Fu et al., 2023). Darker regions indicate areas where the model retains more localized information for those specific groups.

$N$ visual tokens into $M$ tokens, where $M < N$, by grouping neighboring tokens. As shown in Figure 1, the group-wise visual tokens in the LLaVA-1.5-7B model demonstrate varying levels of performance on the MMStar (Chen et al., 2024a) and MME (Fu et al., 2023), suggesting that each token captures localized visual information rather than global concept understanding. Additionally, the central visual tokens contain more informative content compared to those at the edges.

**LLVMs are permutation invariant in terms of visual patch tokens , depending on the benchmark.** From our above results, we empirically verify that each visual patch token from the projector contains localized visual information. Here, we aim to understand how LLVMs systematically process and perceive images based on these visual patch tokens. Given that LLVMs generate answers in an autoregressive manner, we investigate whether they exhibit *order bias* regarding visual patch tokens. To study this, we strongly hypothesize that if LLVMs have *permutation variance*, the performance drop ($\Delta$) will be significant when a random permutation is applied to the visual patch tokens $\mathbf{X}_V$.

As shown in Table 1, the overall performance across most benchmarks declines when the visual patch tokens are randomly shuffled. However, the performance gap between the original and the shuffled (Perm.) versions is not substantial, remaining within a 0–2% range, for LLaVA-1.5 and LLaVA-NeXT. Considering that LLaVA-1.5 uses 576 visual tokens, this is an intriguing observation. It suggests that current LLVMs interpret images in a *global* manner, despite each visual patch token containing localized information (see Figure 1), and even though they process both images and text autoregressively. In the case of LLaVA-OneVision which has many visual tokens (729), the avg. performance drop ($\Delta$) is non-trivial. Upon closer analysis, we find that the "permutation invariance"

Figure 3: We present examples of shuffled images with different grid sizes (2, 4, 8, 14) derived from a MathVista dataset image. As the grid size increases, the chart image becomes more artistically styled.

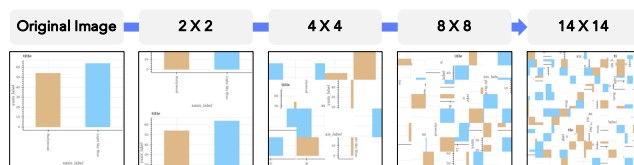

property is both benchmark-dependent and capability-dependent. Specifically, in perception-related benchmarks, such as MMVP and Q-Bench, the performance drop is minimal. In fact, for `LLaVA-1.5` and `LLaVA-NeXT`, performance even slightly improves in some cases. On the other hand, in text-rich benchmarks requiring reasoning capabilities (e.g., MathVista and ChartQA), the performance drops significantly. These benchmarks demand an understanding of detailed numerical information and highly structured geometric graphs, where preserving the spatial structure of visual patch tokens is crucial. We hypothesize that this global interpretation may result from recent LLVMs being trained via backpropagation, with the loss signal primarily derived from the text output of the `Assistant:` side. Based on these experiments, we argue that while LLVMs are trained with an autoregressive objective, they internally handle images globally. This observation offers a possible explanation for the success of pixel shuffling (Chen et al., 2024c) in achieving both strong performance and efficiency.

**LLVMs are sensitive to spatial structures.** Instead of treating visual patch tokens as permutation invariants, we explore how LLVMs behave when the sequence of image patches is permuted. To examine the sensitivity to spatial structure, we randomly shuffle image patches at varying grid sizes (2, 4, 8, 14), as shown in Figure 2. In our experiments, we observe that `LLaVA-OneVision` is sensitive to spatial structures on the MathVista (Lu et al., 2023) and AI2D (Kembhavi et al., 2016) tasks, despite the ViT learning all interactions between image patches. This result contrasts with previous study (Naseer et al., 2021) suggesting that ViT-based vision encoders exhibit high permutation invariance to patch positions than CNN counterparts. We posit that on the MMVP Tong et al. (2024) dataset, which involves perception task, `LLaVA-OneVision` would

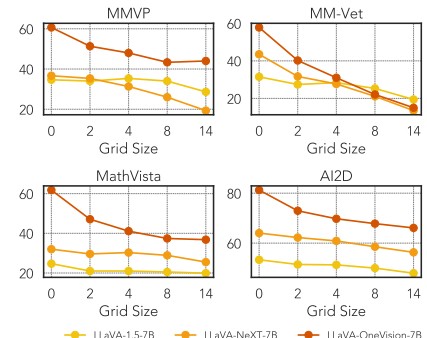

Figure 2: We present the performance across different grid sizes (2, 4, 8, 14) on the MMVP, MM-Vet, MathVista, and AI2D datasets, using three models: `LLaVA-1.5`, `LLaVA-NeXT`, and `LLaVA-OneVision`.

also show permutation invariance to randomly shuffled patches, similar to existing work (Naseer et al., 2021) analyzing the ImageNet (Deng et al., 2009) val. dataset. However, unlike ImageNet, the MathVista and AI2D datasets contain more structurally complex images (e.g., charts, code screenshots) that are highly sensitive to spatial structure, as the original numerical understanding is significantly disrupted. Shuffling image patches in such cases disrupts geometric relationships or relative magnitudes in plots or charts, making accurate interpretation of these images significantly more challenging. Interestingly, both `LLaVA-1.5` and `LLaVA-NeXT` exhibit insensitivity to spatial structure, particularly in the MathVista dataset, where performance drops were minimal. These results suggest the need for further investigation, which we address in the following sections.

### 3.4 DO LLVMS PERCEIVE NUMERICAL INFORMATION WELL?

Here, we study whether LLVMs truly perceive text-rich images (e.g., charts, geometric shapes) that contain highly detailed numerical and shape information. To do this, we construct synthetic datasets for MMVP (Tong et al., 2024) and MathVista (Lu et al., 2023). Specifically, we first generate an image description of a given image using `LLaVA-OneVision` (Li et al., 2024b) with the prompt: "*Please generate a caption of this image.*". Next, we generate an image corresponding to the image description leveraging the `SDXL-Lightning` (Lin et al., 2024) model, ensuring both quality and efficiency. As a result, we get synthesized version (`Syn.`) to the original version (`Org.`), illustrated in Figure 5.

Figure 5: We present examples of images (left) synthesized by `SDXL-Lightning` and (right) occluded using three methods: `Random`, `Salient`, and `Non-Salient`. The original images are from the Math-Vista and MME datasets. Occluded areas are marked in black to indicate zero pixel values.

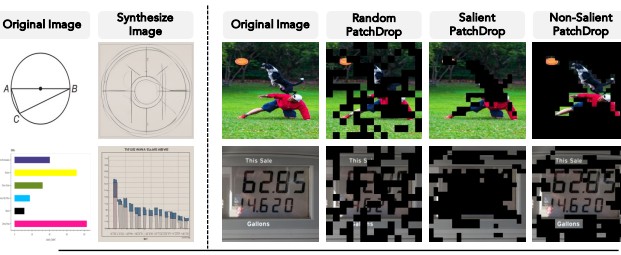

**In some cases, LLVMs can solve problems without seeing the image.** Table 2 presents the performance comparison on both original and synthesized datasets. For comparison, we evaluate the knowledge-embedded-specific LLVM, `Meteor` 7B (Lee et al., 2024c). Overall, compared to the basic perception task (i.e., MMVP), the performance drop in Math-Vista is not significantly larger across four LLVMs. Given that the generated images show distorted chart and function shapes, with detailed numerical and formula information missing, as shown in Figure 5 (`CLIP-I` scores lower than in MMVP), it is surprising that LLVMs are still able to solve math problems requiring advanced cognitive reasoning, without key information.

|  |  |  | MMVP | | MathVista | | |
|---|---|---|---|---|---|---|---|
| LLVMs | Text-rich? | Scale | Orig. | Syn. | Orig. | Syn. | Freq. |
| CLIP-I | - | - | - | 0.84 | - | 0.61 | - |
| LLaVA-1.5 | ✗ | 158K | 34.7 | 20.7 (▼ 14.0) | 24.7 | 22.9 (▼ 1.8) | 81.0 |
| LLaVA-NeXT | ✓ | 760K | 36.7 | 16.7 (▼ 20.0) | 32.0 | 27.7 (▼ 4.3) | 50.0 |
| Meteor | ✓ | 1.1M | 51.3 | 35.3 (▼ 16.0) | 52.1 | 31.4 (▼ 20.7) | 9.5 |
| LLaVA-OneVision | ✓ | 3.1M | 60.7 | 37.3 (▼ 23.3) | 61.8 | 37.0 (▼ 24.8) | 12.0 |

Table 2: We present the performance on the synthesized versions of the MMVP (Tong et al., 2024) and Math-Vista (Lu et al., 2023) datasets across the models `LLaVA-1.5`, `LLaVA-NeXT`, `Meteor`, and `LLaVA-OneVision`. Additionally, we provide the scale of the visual instruction training datasets used by each model and specify whether chart, math, and diagram datasets were included. `CLIP-I` indicates the image similarity using `CLIP-ViT-L/14`. Freq. denotes the frequency with which the model generates the answer "1" in free-form question types in `Syn. cases.`

This observation leads us to more in-depth analysis on MathVista dataset. We analyze how LLVMs solve math problems using synthesized images. In instances where they answer correctly, LLVMs frequently choose "*No*" for MCQs and tend to generate "*1*" for free-form responses. A deeper analysis reveals that many of these questions ask "*What is the smallest value?*", causing the models to select "1" using commonsense reasoning, without needing to interpret the image itself. Table 2 shows how often the models produce "1," with a noticeable drop in frequency for `LLaVA-OneVision` and `Meteor` models. This suggests that these models, likely due to extensive training with million-scale datasets, struggle with "smallest value" questions when images are unclear, demonstrating their ability to interpret images effectively.

**Scaling up visual instruction tuning datasets improves text-only mathematical reasoning.** Here, we explore whether enhancing math reasoning in a visual context can improve standard text-only math reasoning. We evaluate four LLVMs on the GSM8K (Cobbe et al., 2021) dataset in an 8-shot setting using Chain-of-Thought (CoT) prompting (Wei et al., 2022). As shown in Figure 4, we observe that models performing well in visual math contexts also achieve strong performance on GSM8K. Moreover, as the scale of the dataset used for training increases, so does model performance. This suggests that using high-quality, large-scale datasets (e.g., rationale-style datasets, as used in `Meteor`) is beneficial, and that there is compatibility between visual math and text-only math reasoning, aligning with the data-centric AI perspective (Xu et al., 2023; Zhou et al., 2024).

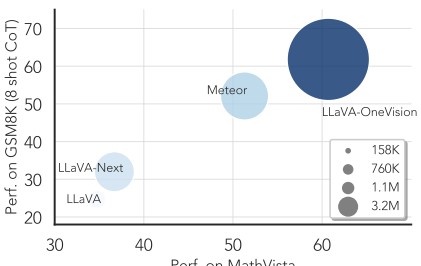

Figure 4: We present performance on the GSM8K dataset using 8-shot Chain-of-Thought prompting. Additionally, we demonstrate that scaling up the instruction-tuning dataset enables LLVMs to solve text-only math reasoning problems more effectively.

### 3.5 ARE LLVMS ROBUST TO OCCLUSIONS?

Existing studies (Naseer et al., 2021; Vishniakov et al., 2023) have demonstrated that ViT models exhibit a remarkable degree of robustness to occlusions, such as patch dropping, than CNN counterparts. Since most LVLMs utilize `CLIP ViT-L` as their vision encoder, we aim to explore

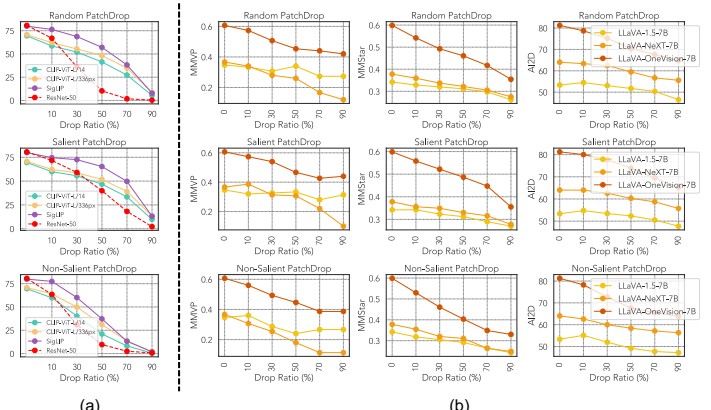

Figure 6: We present robustness performance under occlusion conditions. (a) ViT variant vision encoders demonstrate greater robustness to occlusion compared to `ResNet-50`. (b) LLVMs also show robustness to occlusion, benefiting from the use of ViT encoders.

whether this robustness transfers to LVLMs in scenarios involving occluded images. Following the simple masking method presented in prior work (Naseer et al., 2021), we manipulate the input image $I = \{x_i\}_{i=1}^{N}$, where $N$ represents the number of patches. Specifically, we mask out $N'$ patches (where $N' < N$) by setting the pixel values of these patches to zero, creating an occluded image, denoted as $\tilde{I}$. We then apply three distinct occlusion methods to the image $I$: (1) `Random PatchDrop`: A subset of $N'$ patches is randomly selected and dropped from the image, effectively simulating random occlusion; (2) `Salient PatchDrop`: We strategically select and drop salient patches by leveraging the self-supervised ViT model `dino-small` (Caron et al., 2021); (3) `Non-Salient PatchDrop`: In this case, we drop non-salient, background patches, retaining only the salient information. This method also utilizes `dino-small`, following a similar approach to the `Salient PatchDrop` but focusing on removing the background regions. Figure 5 presents example images with different occlusion methods applied.

**LLVMs are robust against occlusion.** Before evaluating LLVMs on occluded images, we first verify whether ViT-based encoders are more robust than their CNN counterparts in this scenario. To do this, we assess several ViT variants and `ResNet-50` (He et al., 2016) on occluded ImageNet (Krizhevsky et al., 2009) images, applying the same masking process as mentioned above. As shown in Figure 6 (left), compared to `ResNet-50`, ViT variants demonstrate greater robustness in occlusion scenarios, consistent with findings from the prior study (Naseer et al., 2021). Due to this robustness, LLVMs also exhibit relatively strong performance under occlusion. This result is surprising given that in the AI2D dataset, which contains text-rich diagram images, 50%–70% of the patches are missing, yet LLVMs can still provide correct answers to some extent. This may be because AI2D involves selecting one answer from multiple options, suggesting the possibility of selection bias (Zheng et al., 2023a), a significant issue that we leave for future work.

## 3.6 DO LLVMs PRESERVE CROSS-MODAL ALIGNMENT?

In the *de facto* structure of LLVMs, a projector $f_P$ enables LLMs to perceive and understand images by transforming visual representations into the LLM's representation space. While a recent work (McKinzie et al., 2024) suggests that the type of projector has minimal impact on performance, other studies (Zhai et al., 2024; Verma et al., 2024) have argued that the projector have limitations in preserving cross-modal understanding and issues such as catastrophic forgetting. In this work, we investigate (1) how effectively a trained projector maintains its *visual recognition* capability relative to the LLVM's original vision encoders (e.g., `CLIP-ViT-14` for `LLaVA-NeXT`), and (2) how well a trained projector preserves cross-modal alignment, based on the *platonic representation hypothesis* (Huh et al., 2024), compared to representation expressivity without alignment learning.

**LLVMs struggle to preserve the original visual understanding capability.** Ideally, after alignment and visual instruction tuning, LLVMs should retain the visual perception abilities of their original vision encoders, allowing them to understand and classify images effectively. To assess this, we evaluate LLVMs on zero-shot image classification tasks using widely adopted datasets such as Caltech100 (Higgins et al., 2017), Food101 (Bossard et al., 2014), CIFAR-100 (Krizhevsky

| LLVMs | Caltech101 | CIFAR-100 | Food101 | Pets | Country211 | EuroSAT | AirCraft | Avg. |
|---|---|---|---|---|---|---|---|---|
| CLIP ViT 336 | 84.50 (0.52) | 75.10 (1.74) | 93.72 (0.61) | 93.48 (0.63) | 31.14 (0.84) | 58.90 (0.84) | 34.08 (0.77) | 67.27 (0.85) |
| LLaVA-1.5 | 43.76 (2.69) | 48.36 (2.47) | 57.22 (0.61) | 73.22 (0.37) | 12.20 (0.47) | 11.72 (0.34) | 17.00 (0.72) | 37.64 (1.10) |
| | (▼ 40.74) | (▼ 26.74) | (▼ 36.5) | (▼ 20.26) | (▼ 18.94) | (▼ 47.18) | (▼ 17.08) | (▼ 29.63) |
| CLIP ViT 14 | 84.52 (0.56) | 75.86 (1.06) | 92.78 (0.41) | 93.08 (0.30) | 28.68 (1.44) | 58.46 (0.80) | 32.98 (1.02) | 66.62 (0.80) |
| LLaVA-NeXT | 56.68 (2.29) | 45.36 (1.02) | 53.14 (1.00) | 75.06 (0.93) | 12.94 (0.35) | 8.34 (0.59) | 12.66 (0.46) | 37.74 |
| | (▼ 27.84) | (▼ 30.5) | (▼ 39.64) | (▼ 18.02) | (▼ 15.74) | (▼ 50.12) | (▼ 20.32) | (▼ 28.88) |

Table 3: We report the Top-1 accuracy (%) with standard deviation (in parentheses) of LLVMs and their corresponding vision encoder models on 1K subsampled datasets from Caltech100, CIFAR-100, Food101, Pets, Country211, EuroSAT, and AirCraft. We run five experiments.

et al., 2009), Pets (Parkhi et al., 2012), Country211 [2], EuroSAT (Helber et al., 2019), and Air-Craft (Maji et al., 2013). Following the method in previous work (Zhai et al., 2024), we use ChatGPT (gpt-3.5-turbo) (OpenAI, 2023) to extract a single label with the use of prompt: *Is this prediction correct?*. As shown in Table 3, the performance of LLVMs significantly degrades across all datasets compared to their vision encoders, suggesting that LLVMs do not fully retain the perception capabilities of their original vision encoders. This may be due to: (1) LLVMs being trained to solve complex tasks (e.g., chart or math reasoning) with the use of instruction, which may cause them to lose basic perception abilities (e.g., recognizing simple objects), a phenomenon known as *catastrophic forgetting* (Zhai et al., 2024) [3], and (2) the vision encoder's relatively small parameter size (307M for CLIP ViT-L/336px) compared to the LLM (7B for Vicuna-1.5), which could result in a loss of visual perception capability during projection, as the more powerful LLM dominates.

**LLVMs lose the ability to understand and interpret shared world concepts.** Beyond visual recognition capabilities, we analyze cross-modal alignment based on the *platonic representation hypothesis* (Huh et al., 2024), which argues that neural networks, despite being trained on different objectives, data, and modalities, should converge to a shared statistical model of reality in their representation spaces. To measure representation similarity between two modalities, the original authors of this hypothesis use mutual nearest-neighbor alignment metrics, a type of kernel-alignment metric. In our work, we assess how much alignment is lost after visual instruction tuning by applying this metric within the context of the platonic representation hypothesis. We evaluate 10 LLMs and measure alignment between these LLMs and vision encoders (LLVMs) using the DOCCI (Onoe et al., 2024) dataset which contains long image descriptions requiring localized scene understanding. As shown in Figure 7, after vi-

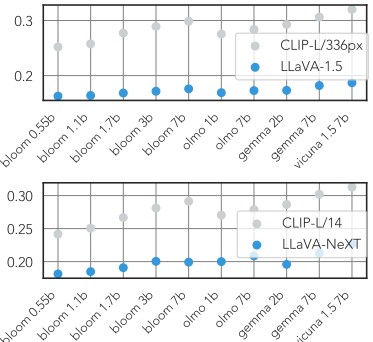

Figure 7: We present how alignment preservation changes (CLIP → LLaVA) in the representation space across various LLM families, BLOOM (Le Scao et al., 2023), OLMo (Groeneveld et al., 2024), Gemma (Team et al., 2024), Vicuna (Chiang et al., 2023), with different parameter sizes on the DOCCI dataset.

sual instruction tuning, both LLaVA-1.5 and LLaVA-NeXT show degraded alignment performance with respect to representations compared to their original vision encoder. This suggests doubts about the actual role of the projector in causing the degradation in alignment preservation. From this observation, we speculate that current LLVMs are trained on a variety of datasets to achieve generalization (i.e., multi-task learning). However, during visual instruction tuning, the models might overemphasize capabilities requiring complex cognition while potentially reducing representations related to other tasks, such as localized scene understanding (i.e., DOCCI). This results in a lower alignment score and catastrophic forgetting, as shown in Table 3. For future work, one potential direction is to develop a localized enhanced alignment module similar to HoneyBee (Cha et al., 2024).

---

[2] https://github.com/openai/CLIP/blob/main/data/country211.md

[3] On the CLEVR/Count (Johnson et al., 2017) dataset, we observed a 16.6% performance improvement in the LLaVA-NeXT model compared to the previous vision encoder (i.e., CLIP-ViT-L/14.)

### 3.7 MODEL BEHAVIOR: WHICH MODALITY AND LAYERS ARE MORE IMPORTANT?

Here, we conduct an in-depth analysis of model behavior to assess the importance of either a layer or a visual token when performing downstream tasks. We hypothesize that if adding arbitrary noise to a specific component—either a layer block or a visual token—results in a significant drop in model performance, then that component is crucial and actively involved in the model's reasoning process. To quantify this, we define an *importance* score ($\mathcal{I}$) inspired by the concept of "*sharpness of minima*" (Keskar et al., 2016; Lee et al., 2024f). This concept aims to identify flat minima, which promote stable training and better generalization, by measuring the sensitivity of the training function around a local minimum. In our work, we adapt this concept for the inference stage.

**Definition 3.1** (Importance Score). Let $x_t \in \mathbb{R}^d$ be the $d$-dimensional input embedding for a target subject $t$. For $x_t$, we define the constraint candidate set $\mathcal{C}_t$ for $t$ as:

$$\mathcal{C}_t = \{z_t \in \mathbb{R}^d : -\epsilon + |x_t| \le z_t \le \epsilon + |x_t|\}, \quad \epsilon \sim \text{Uniform}(-1, 1), \tag{2}$$

where $z_t$ is a noise vector. The importance score $\mathcal{I}$ for target $t$ is then defined as:

$$\mathcal{I}_t := \frac{f(x_t) - \max_{z_t \in \mathcal{C}_t} f(x_t + z_t)}{f(x_t)} \times 100. \tag{3}$$

Note that while the concept of "sharpness of minima" was originally used to find flat minima during training by defining a square-bound constraint set, this is feasible because the model is trained via stochastic gradient descent (SGD), which indirectly allows the evaluation of all noise values $z$ in the constraint set $C$. However, since our experiment focuses on downstream task performance during inference, we adapt this concept by sampling $K$ noise vectors, $\{z_t^1, z_t^2, \ldots, z_t^K\} \sim \mathcal{C}_t$, with different random seeds. For computational efficiency, we set $K = 10$.

**LLVMs strongly focus on the center of the image.** To assess the extent to which LLVM utilizes visual token information, we add a noise vector $z_t$ to each visual token information based on the importance score ($\mathcal{I}$). However, performing computations on each individual visual token is computationally intensive, especially for models such as `LLaVA-1.5-7B` that process 576 visual tokens arranged in a $24 \times 24$ grid of patches. To accelerate computation and reduce complexity, we adopt the same process as 3.3. As shown in Figure 8, `LLaVA-1.5-7B` places strong emphasis on the central part of the images in the MM-Vet dataset, while the edge regions have minimal influence on the final performance compared to the central visual tokens. This result suggests that not all visual tokens are necessary, aligning with recent works (Cha et al., 2024; Xue et al., 2024) that reduce redundant visual tokens in the projector to enhance efficiency.

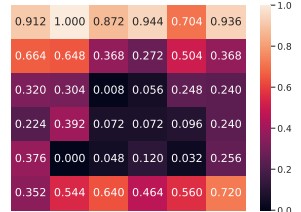

Figure 8: We report the degree of utility of group-wise visual tokens for LLaVA 1.5 7B on the MM-Vet dataset. Darker regions indicate that the LLVM relies heavily on information from those specific group parts.

**Lower block layers in LLVMs are more important.** Figure 9 (left) shows that the lower layers ($< 6$) play a crucial role in handling the integrated capabilities required for tasks in the MMStar dataset. This suggests that these layers contain more beneficial representations for perception and cognition. This finding aligns with recent work on LLVMs, specifically TroL (Lee et al., 2024b), which introduces the concept of "*layer traversal*." This technique revisits layer

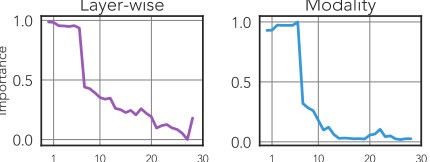

Figure 9: We present the results of (left) layer-wise importance and (right) modality importance within the layers.

representations, resulting in a highly generalizable model despite its small size (1.8B parameters). In their paper (Figure 6), the traversal pattern is more pronounced in the lower layers, which is consistent with our findings. Therefore, we believe our results may provide insights into why traversing layers leads to improved generalization.

**Textual modality is more important than visual modality.** In addition to layer-wise importance, we measure *modality importance* using the score $\frac{\mathcal{I}_I}{\mathcal{I}_T}$, which calculates the relative importance of textual and image modalities. Specifically, to obtain the image modality importance score $\mathcal{I}_I$, we feed

noise vectors to the positions corresponding to image tokens (e.g., 576 tokens in `LLaVA-1.5`), and vice versa for text modality $\mathcal{I}_T$. As shown in Figure 9 (right), until the lower layers ($< 8$), the image modality is more important than the textual modality. However, as layers progress, we observe that the textual modality becomes increasingly important, likely because generating responses requires a stronger focus on text with the perspective of autoregressive modeling. This suggests that LLVMs initially focus on global image perception (section 3.3), and by the middle layers, they have processed the image and shifted toward solving complex user queries to generate coherent answers. Similarly, in TroL, layer traversal occurs more actively in the lower layers, which we interpret as the model attempting to better comprehend the image when it fails to do so in a single pass, enabling it to solve complex reasoning tasks more effectively. These findings highlight the value of strong visual perception, which may explain the success of models utilizing large visual tokens (Wang et al., 2024; Li et al., 2024b) or high-resolution image processing (Li et al., 2024b).

## 4 DISCUSSIONS

**Building more interactive evaluation benchmarks.** As mentioned in section 3.4, LLVM can effectively solve problems even without seeing the input image. However, current evaluation benchmarks are designed for single-turn interactions and lack applicability to real-world, interactive scenarios. For example, in standard OCR tasks, we typically assess whether the LLVM correctly transcribes text from an image. But consider a practical situation: you're traveling in a foreign country and visiting a local restaurant. Translating the menu is challenging, and while an application with strong OCR capabilities would be helpful, this is only the first step. When ordering, the LLVM should not only recognize the menu items but also understand the user's preferences — what they like or dislike — by incorporating knowledge of their persona (Lee et al., 2022). Therefore, future benchmarks should be more interactive and socially grounded (Zhou et al., 2023a), extending beyond multiple-choice, binary, or non-interactive free-form tasks. These benchmarks should involve multi-turn interactions and be based on the user's preferences (Lee et al., 2024g) or persona in long-term social interactions (Jang et al., 2023; Lee et al., 2024h).

**A new paradigm enhancing cross-modal alignment.** Current LLVMs have widely adopted the *model-stitching* structure, which demonstrates impressive capabilities on tasks requiring higher-level cognitive reasoning. However, they exhibit significantly degraded performance in zero-shot image classification tasks (Table 3). Additionally, they cannot effectively preserve alignment (Figure 7) in terms of relatively simple perception when compared to text-rich images (e.g., charts, mathematical expressions). Recently, although recent studies (Li et al., 2024b; Lee et al., 2024c) has been extensively scaling up model sizes with larger datasets to achieve higher performance on increasingly difficult tasks — which we believe is the correct direction - we think it is necessary to deeply consider innovative model architectures (e.g., layer of traversal (Lee et al., 2024b), hidden dimension expansion (Lee et al., 2024a)) to enhance cross-modal alignment at least once. For example, in recent unified architectures (Xie et al., 2024; Erfei Cui, 2024), enabling LLMs to generate images is akin to how drawing can be substantially more challenging for humans than simply viewing a picture. This is because drawing requires a comprehensive and simultaneous understanding of complex world concepts such as relationships between objects, lighting, perspective, and more. Therefore, by projecting visual imagination abilities (Lu et al., 2022b; Lee et al., 2023) onto LLVMs to enable them to generate images, it might significantly help in better preserving cross-modal alignment.

## 5 CONCLUSION

In this paper, we systemically reveals intriguing properties of LLVMs with respect to *permutation invariance*, *robustness*, *alignment preserving*, and *importance* under various image settings such as occlusion and synthesized images. We hope these findings will assist academic researchers and ML developers in advancing the next frontier of LLMs by providing a foundational basis for future model design choices.

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

## A LIMITATIONS AND DISCUSSIONS

In this work, we observe intriguing findings regarding LLVMs under various experimental settings. To provide a clear and well-defined scope for our conclusions, we further discuss the limitations of the experimental setup for our findings (or claims), explore the most plausible application directions based on our findings, and offer meaningful insights for future research directions for each finding.

### A.1 LIMITATIONS

Overall, our experiments have several limitations regarding model- and dataset-side generalizability, which are important for a more rigorous analysis. For instance, we primarily evaluate LLVMs on VQA-style tasks, including free-form and multiple-choice question types, and focus exclusively on the LLaVA family. To improve the generalizability of our findings, future work should explore experiments on other LLVMs, such as `Qwen2-VL` Wang et al. (2024), and extend evaluations to additional datasets (e.g., image captioning datasets). Furthermore, demonstrating the impact of model scaling would provide stronger support for our conclusions. Below, we present the specific limitations for each section.

**Limitations: Section 3.3.** In Figure 1, obtaining the results required running computations for the full number of visual patch tokens, which is highly resource-intensive. This is especially challenging given the large number of visual patch tokens required by recent LLVMs—for example, 576 for `LLaVA-1.5` and more than 1,000 for `Qwen2-VL` Wang et al. (2024).

**Limitations: Section 3.4.** Synthesized images were generated using `LLaVA-OneVision-7B` Li et al. (2024b) with the prompt template: "*Please generate a caption of this image.*" and `SDXL-Lightning` Lin et al. (2024). To improve robustness, future experiments should explore captions with varying levels of detail, from concise to highly detailed, by using alternative prompt templates, specialized captioning models (e.g., `ShareCaptioner` [4] Chen et al. (2023a)), or more advanced text-to-image generation models that outperform `SDXL-Lightning`. Incorporating these variations would enhance the reliability of our conclusions.

**Limitations: Section 3.5.** During patch-dropping, we employed the `dino-small` Caron et al. (2021) model for both `Salient PatchDrop` and `Non-Salient PatchDrop`. The impact of patch dropping is likely to vary depending on the size and type of self-supervised vision model used (e.g., large-scale DINO), potentially leading to differing patterns of performance degradation.

**Limitations: Section 3.6.** While we evaluated visual perception capabilities across various image datasets, many domain-specific image datasets exist in the real world. To draw more generalizable conclusions, it would be beneficial to evaluate additional datasets, such as the VTAB benchmark Zhai et al. (2019). Additionally, we investigated *catastrophic forgetting* by following existing experimental setups from the prior study Zhai et al. (2024). However, comparing LLVMs with contrastive approaches (e.g., `CLIP`) may be unfair due to multiple factors influencing LLVM performance, such as prompt variations and methods for calculating accuracy from the generated text. To enable a more rigorous analysis, future work should explore different prompt methods and fine-tune LLVMs on zero-shot image classification datasets (e.g., CIFAR-100) to assess whether perception capabilities improve. Regarding the LLM-dominance problem during visual instruction tuning, confirming this phenomenon is challenging. To test it effectively, LLVMs should be trained with identical datasets but varying LLM sizes and vision encoder scales. Alternatively, other types of LLVMs that incorporate external computer-vision models (e.g., segmentation models) such as `MoAI` Lee et al. (2024e) could be evaluated. Using visually enhanced LLVMs would strengthen this argument. In addition, for Figure 7, evaluating cross-modal alignment on a broader variety of datasets, such

---

[4] `https://huggingface.co/Lin-Chen/ShareCaptioner`

as CC12M Changpinyo et al. (2021), WIT Srinivasan et al. (2021), and RedCaps12M Desai et al. (2021), would provide a better understanding of the findings. Expanding this evaluation to various LLVMs, such as `LLaVA-OneVision` and `Qwen2-VL`, would also yield more comprehensive insights.

**Limitations: Section 3.7.** In Figures 8 and 9, obtaining the importance scores is computationally expensive. For a single run, we calculate the importance scores for each group-wise position (e.g., 36 positions for `LLaVA-1.5`), and we repeat the experiment $K$ times (with $K = 10$). This results in a total of 360 experiments per benchmark. Similarly, the computation for layer importance is also resource-intensive.

## A.2 DISCUSSIONS

Here, we present several discussions based on our findings.

**Findings: Permutation Invariance.** We suggest that future work focuses on two key directions. First, it is essential to develop more challenging benchmarks that better explore LLVMs' capabilities. Such benchmarks should prioritize free-form question types and avoid including "blind" samples Fu et al. (2024); Li et al. (2024a) that models can solve using commonsense reasoning without actually perceiving the image. Building multi-turn interactive conversation benchmarks, like MMDU Liu et al. (2024d), could be particularly useful in this context. Second, since LLVMs generally exhibit permutation invariance, visual patch tokens can be treated as independent elements, allowing images to be represented as unordered sets of points. Applying paradigms like "Context Clusters," Ma et al. (2023) which rely on clustering algorithms rather than convolutions or attention mechanisms, could improve interpretability and training efficiency. Furthermore, this approach could facilitate generalization to other data domains, such as point clouds Ma et al. (2022), RGB-D data, or sensory images Yu et al. (2024), broadening the applicability of LLVMs.

**Findings: Sensitivity to Spatial Structures.** One future direction is to develop more robust LLVMs that can handle spatial disruptions. Real-world images often lack perfect clarity—details may be missing, images may be flipped, or other disruptions may occur. To address this, we propose incorporating randomly shuffled images into the training process. By framing this as a jigsaw puzzle Chen et al. (2023b) task, models can be trained to reconstruct the original positions of image patches. This approach could enhance their robustness to spatial variations, making them more applicable to real-world scenarios.

**Findings: Catastrophic Forgetting.** Balancing perception and cognitive reasoning capabilities is critical. The "catastrophic forgetting" problem Kirkpatrick et al. (2017) has been a long-standing issue in machine learning. A standard approach is to train models on mixed datasets Ke et al. (2020); Gururangan et al. (2020) with a carefully designed balance (a "golden ratio") between perception- and reasoning-related data. Continuously training LLVMs on perception-focused datasets following rehearsal methods Rebuffi et al. (2017) can minimize catastrophic forgetting by retaining knowledge of prior tasks while learning new ones. Knowledge distillation Jin et al. (2021) from large-scale LLVMs (e.g., 72B parameters) to smaller-scale models (e.g., 7B parameters) could help preserve perception capabilities while maintaining reasoning strength. Alternatively, fine-tuning adapters (e.g., p-tuning Liu et al. (2021), LoRA Hu et al. (2021), Q-LoRA Dettmers et al. (2024)) on task-specific datasets offers a lightweight solution to improve performance on new tasks without sacrificing existing capabilities.

**Findings: Cross-modal Alignment in the Platonic Representation Hypothesis.** Maintaining the original cross-modal alignment is critically important. Continual learning methods (presented above) could be applied to mitigate the loss of alignment during visual instruction tuning. Enhancing the visual perception capability of the projector during training could also help. For instance, employing models such as HoneyBee Cha et al. (2024), which incorporate convolution layer-based projectors, could improve localized understanding. Convolution layers are well-known for their strong inductive bias toward localized feature extraction, making them better suited for capturing fine-grained details in images. Even with the inclusion of complex instruction datasets (e.g., charts, math), a carefully designed projector that excels at extracting detailed and localized information from images could naturally improve both perception and reasoning capabilities. We hypothesize

that enhancing localized perception would inherently lead to improvements in reasoning, aligning the two capabilities more effectively.

**Findings: Importance of Central Visual Tokens.** Based on our observations, reducing redundant visual tokens in the projector could enhance training and inference efficiency, aligning with findings from prior studies Alayrac et al. (2022); Cha et al. (2024); Xue et al. (2024). Typically, the large number of visual tokens poses a computational burden. This is particularly relevant for real-world scenarios where interleaved format-style conversations Li et al. (2024c); Lee et al. (2024h) are predominant. High visual token counts can make it challenging to train more effective LLVMs for such interleaved conversational formats. Our findings provide a practical direction for reducing visual token counts while maintaining performance. By doing so, we can enable the training of interleaved-format LLVM models more efficiently, similar to approaches highlighted in previous research Xue et al. (2024).

**Findings: Importance of Lower Layer.** Based on our observations, we emphasize the importance of the traversing layers (TroL) approach Lee et al. (2024b), in improving generalization. In this approach, models are trained to revisit and leverage layer-specific information during the training process. The paper demonstrates that lower layers are more actively engaged, which aligns with our findings. These results suggest that the lower layers of LLVMs play a critical role in establishing a foundational understanding of the world. To enhance this capability, increasing the signal for world understanding in the lower layers during training could be a promising direction. One potential method is injecting noise information into the lower layers during training, as suggested in a prior study Jain et al. (2023). This technique could improve the robustness of LLVMs, further solidifying their foundational perception and reasoning capabilities.

**Findings: Relative Importance of Modalities.** While the textual modality appears more influential in higher layers, improving the visual perception capability in lower layers is crucial. This is because LLVMs rely heavily on understanding the given image during the initial processing stages. As suggested in prior works Cha et al. (2024); McKinzie et al. (2024), using a larger number of visual tokens, adopting high-resolution image processing Li et al. (2024c), or employing dynamic image processing methods Wang et al. (2024); Li et al. (2024b) is essential for enhancing performance. Furthermore, strengthening the projector's capability for localized visual understanding Cha et al. (2024) could be beneficial. For instance, after the initial image-caption alignment step (commonly the first step in LLVM training), an additional training phase called "empowering localized understanding" could be introduced before visual instruction tuning. This phase would involve adding an extra layer, referred to as the "AL" (Augmented Layer), on top of the simple linear layer. The AL would be trained using a masked autoencoder (MAE) approach He et al. (2022), where the model learns to predict masked image patches. This process would enhance localized visual understanding, ultimately improving the balance between visual and textual modalities and boosting overall model performance.

## B ADDITIONAL EXPLANATION OF PLATONIC REPRESENTATION HYPOTHESIS

In Section 3.6, we investigate how effectively a trained projector preserves cross-modal alignment, drawing on the *Platonic Representation Hypothesis* Huh et al. (2024). In this section, we provide a detailed explanation of (1) the definition of the Platonic Representation Hypothesis, (2) the alignment metric, and (3) the methodology used to measure alignment in our experiment.

### B.1 DEFINITION OF THE PLATONIC REPRESENTATION HYPOTHESIS

Traditionally, different types of AI models represent the world in fundamentally different ways. For instance, when presented with the same reality (e.g., an image, as illustrated in Figure 10), self-supervised vision models might focus on shapes, colors, and optical effects — features critical to visual understanding — while LLMs might emphasize semantic meanings and syntactic structures. Recently, researchers have developed LLVMs by jointly training vision models and LLMs, encouraging them to interpret and represent the world in a more unified manner. The Platonic Representation Hypothesis posits that neural networks, trained with distinct objectives on different data

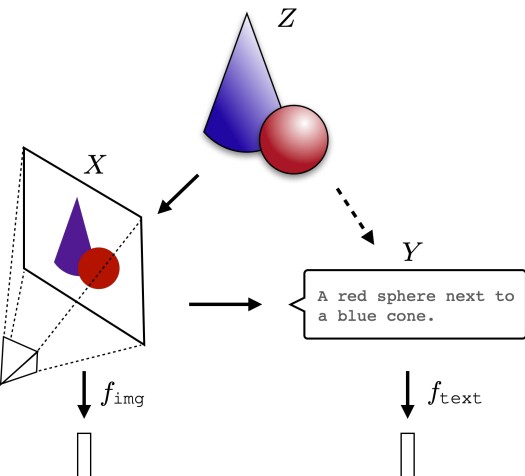

Figure 10: Images ($X$) and text ($Y$) are projections of a common underlying reality ($Z$). We conjecture that representation learning algorithms will converge on a shared representation of $Z$, and scaling model size, as well as data and task diversity, drives this convergence. For clarity, this figure and its caption have been taken exactly as they appear in the original paper Huh et al. (2024).

modalities, converge toward a shared statistical model of reality in their representation spaces. In the original paper introducing this hypothesis, the authors demonstrated a strong level of alignment between the representations of models trained on disparate modalities (e.g., Figure 3 in the original paper). Based on these findings, we argue that the alignment between models trained on different modalities should not only be preserved but potentially strengthened.

### B.2 ALIGNMENT MEASUREMENT.

To evaluate the alignment between representations from two models, we employ the **Mutual $k$-Nearest Neighbor (MNN) Metric**. This metric focuses on local similarity by computing the intersection of the $k$-nearest neighbor sets for each sample from the two models' representation spaces. The alignment is then measured based on the size of these intersections, as detailed below.

**Mutual $k$-Nearest Neighbor Metric.** Let $f$ and $g$ denote the representation functions of two models, and let $\mathcal{X}$ represent the data distribution (e.g., an image-caption dataset).

1. The representations for a mini-batch of samples $\{x_i, y_i\}_{i=1}^{b}$ are defined as:
$$\phi_i = f(x_i), \quad \psi_i = g(y_i), \quad i = 1, \ldots, b,$$
where $\Phi = \{\phi_1, \ldots, \phi_b\}$ and $\Psi = \{\psi_1, \ldots, \psi_b\}$ represent the feature sets produced by models $f$ and $g$, respectively.

2. For each feature $\phi_i$ and $\psi_i$, the $k$-nearest neighbor sets are computed as:
$$\mathcal{S}(\phi_i) = \{k \text{ nearest neighbors of } \phi_i\}, \quad \mathcal{S}(\psi_i) = \{k \text{ nearest neighbors of } \psi_i\}.$$

3. The alignment for a given pair of features $(\phi_i, \psi_i)$ is defined as the normalized size of the intersection of their $k$-nearest neighbor sets:
$$m_{\text{NN}}(\phi_i, \psi_i) = \frac{1}{k} |\mathcal{S}(\phi_i) \cap \mathcal{S}(\psi_i)|,$$
where $|\cdot|$ represents the size of the intersection.

4. The overall alignment for the mini-batch is computed as the average alignment across all samples:
$$M_{\text{NN}} = \frac{1}{b} \sum_{i=1}^{b} m_{\text{NN}}(\phi_i, \psi_i).$$

### B.3 HOW TO MEASURE IN OUR EXPERIMENT

To assess the alignment between a suite of large language models (LLMs) and vision models, we utilize the image-caption pair dataset DOCCI Onoe et al. (2024). Specifically, in DOCCI, the dataset consists of image-caption pairs

$$D = \{(x_i, y_i)\}_{i=1}^{|D|},$$

where $x_i$ denotes the image and $y_i$ denotes the corresponding caption text.

For our experiment, we prepare three models: an LLM ($f_L$), a vision encoder from a vision-language model without visual instruction tuning ($f_V$), and a vision encoder with a projector, representing a vision-language model with visual instruction tuning ($f_{VP}$). The vision encoder in $f_{VP}$ is kept identical to $f_V$. For example, CLIP-L/336px is used as the vision encoder for both $f_V$ and $f_{VP}$ when paired with LLaVA-1.5.

In our experiment, we explore the degree of alignment lost after visual instruction tuning, guided by the Platonic representation hypothesis. We assume that in a successful LLVM, the projector should effectively represent the visual world and enable the LLM to understand and interpret the given image accurately. We calculate two alignment scores: one between $f_L$ and $f_V$, and another between $f_L$ and $f_{VP}$. The discrepancy between these scores reflects the extent to which alignment performance deteriorates.

To compute the alignment scores, we follow these steps:

1. Extract features from $f_L$ by providing the input text $y_i$. We then apply average pooling to all the extracted hidden states.

2. Extract features from $f_V$ by providing the image $x_i$, using only the feature corresponding to the [CLS] token.

3. Extract features from $f_{VP}$ by providing the image $x_i$, applying average pooling to all visual patch tokens (e.g., 576 tokens in LLaVA-1.5) produced by the projector.

Finally, we calculate the alignment scores using these extracted features via the mutual nearest-neighbor alignment metric.

### B.4 MOTIVATION BEHIND SELECTING THE DOCCI DATASET

We posit that the ability to perceive and reason based on complex images (e.g., charts, mathematical representations, code snippets, and diagrams) is crucial for creating a helpful assistant. However, we believe that an LLVM must first excel at understanding more natural scenes to become a broadly applicable personal AI assistant, such as one integrated into smart glasses (e.g., Meta AI's glasses[5]) or real-time cameras (e.g., Project Astra [6]). To achieve effective alignment between the language and vision modalities, we require paired datasets where the captions provide detailed descriptions of the corresponding images. These descriptions must include essential visual features such as attributes, spatial relationships, object counts, objects, text rendering, viewpoints, optical effects, and world knowledge. Based on this criterion, we sought an image-caption pair dataset emphasizing (1) natural scenes and (2) highly descriptive captions. The DOCCI dataset meets these requirements effectively. Of course, other datasets could also be considered as candidates, such as Localized Narratives Pont-Tuset et al. (2020), CC12M Changpinyo et al. (2021), COCO-Caption Lin et al. (2014), WIT Srinivasan et al. (2021), or RedCaps12M Desai et al. (2021). In future work, we plan to conduct additional experiments to enhance the generalizability of our observations.

## C ADDITIONAL EXPLANATION OF IMPORTANCE SCORE

In Section 3.7, we investigate the model's behavior to assess the importance of either a specific layer or a visual token when performing downstream tasks. We hypothesize that introducing arbitrary noise to a specific component — either a layer block or a visual token — will significantly drop the model's performance if that component is crucial to the reasoning process. To quantify this,

---

[5]https://www.meta.com/smart-glasses/

[6]https://www.youtube.com/watch?v=nXVvvRhiGjI

we define an *importance score* ($\mathcal{I}$), inspired by the concept of "sharpness of minima." This section provides a detailed explanation of how the importance score is computed.

**How is Arbitrary Noise Introduced into Target Layers or Visual Tokens?** Based on Equation (2), we prepare the constraint candidate set $\mathcal{C}_t$, defined as a squared boundary:

$$-\epsilon + |x_t| \leq z_t \leq \epsilon + |x_t|, \tag{4}$$

where $\epsilon \sim \text{Uniform}(-1, 1)$. At each iteration, we randomly sample a noise vector $z_t$ and apply it to the target component. Below, we detail how this is done for visual tokens, layers, and modalities.

1. **Visual Token Importance:** When evaluating the importance of a visual patch token (Figure 8), the noise vector is injected into the group-wise visual patch token embeddings at the target position. For instance, Figure 8 illustrates 36 positions. To measure the importance of position 0, we add the noise vector to the corresponding visual patch token embeddings at position 0, while leaving all other patch token embeddings unchanged. These modified embeddings are then input into the LLM for further processing.

2. **Layer-Wise Importance:** To explore layer-wise importance, the noise vector is injected into the target layer before it is processed by the LLM. Specifically, the noise is applied directly to the layer's input embeddings before passing the target layer, ensuring that the perturbation affects only the selected layer.

3. **Modality Importance:** To calculate the importance of the textual modality ($\mathcal{I}_T$), the noise vector is injected only into the positions corresponding to text inputs within the target layer, while leaving the positions associated with visual patch tokens unchanged. Conversely, for visual modality importance ($\mathcal{I}_I$), the noise vector is injected into the positions corresponding to visual patch tokens within the target layer. The relative importance score for each modality is then computed as $\frac{\mathcal{I}_I}{\mathcal{I}_T}$.

To enable better interpretation across layers, all importance scores (both layer-wise and modality-specific) are normalized using `min-max normalization`.

## D  ADDITIONAL EXPLANATION OF EXPERIMENTAL SETUP

In this section, we provide a more detailed explanation of the experimental setup used to obtain our findings, including the required models, preparation of corrupted images, and other specifics. All experiments were conducted using eight A100 GPUs (40GB).

**Experimental Setup: Section 3.3.** We prepared ViT-variant vision encoder-equipped LVLMs that incorporate visual patch tokens. The experiments focus on visual patch tokens processed after the projector. Before conducting the "permutation invariance" experiments, we first demonstrated whether each visual patch token contains localized information. For the experiment on "sensitivity to spatial structure," shuffled images were used, as shown in Figure 2, following the methodology of a prior study Naseer et al. (2021).

**Experimental Setup: Section 3.4.** To generate synthesized images, we utilized an image captioner (`llava-hf/llava-onevision-qwen2-7b-ov-hf`) combined with a text-to-image generative model (`sdxl_lightning_8step_unet.safetensors`). Additionally, a prompt template was carefully designed for this purpose.

**Experimental Setup: Section 3.5.** We prepared occluded images using three masking methods as described in prior work Naseer et al. (2021): `Random PatchDrop`, `Salient PatchDrop`, and `Non-Salient PatchDrop`. To implement `Salient PatchDrop` and `Non-Salient PatchDrop`, we employed the `dino-small` model Caron et al. (2021). Furthermore, to evaluate the robustness of LVLMs to occlusion, we first verified whether ViT-variant encoders exhibit genuine robustness to occlusion by comparing them with CNN-based counterparts, such as `ResNet`.

| LLVMs | MMVP | Q-Bench | MME | MMStar | MM-Vet | LLaVA$^W$ | MathVista | SQA$^I$ | ChartQA | AI2D | Avg. $\Delta$ |
|---|---|---|---|---|---|---|---|---|---|---|---|
| LLaVA-1.5 | 34.67 | 59.73 | 1850.07 | 34.20 | 31.50 | 67.50 | 24.70 | 65.59 | 16.92 | 53.34 | |
| + Perm. | 36.00 | 59.60 | 1874.60 | 33.33 | 30.40 | 66.20 | 21.20 | 65.44 | 14.08 | 52.69 | ▼ 0.59 |
| | (▲ 1.33) | (▼ 0.13) | (▲ 24.53) | (▼ 0.87) | (▼ 1.10) | (▼ 1.30) | (▼ 3.50) | (▼ 0.15) | (▼ 2.84) | (▼ 0.65) | |
| LLaVA-NeXT | 36.67 | 63.55 | 1874.42 | 37.80 | 43.50 | 75.50 | 32.00 | 62.12 | 66.06 | 64.02 | |
| + Perm. | 37.33 | 62.54 | 1890.19 | 36.87 | 43.40 | 75.80 | 21.70 | 62.12 | 34.55 | 64.02 | ▼ 2.71 |
| | (▲ 0.67) | (▼ 1.00) | (▲ 15.78) | (▼ 0.93) | (▼ 0.10) | (▲ 0.30) | (▼ 10.30) | (▼ 0.00) | (▼ 31.51) | (▼ 0.00) | |
| LLaVA-OneVision | 60.67 | 77.26 | 1982.5 | 59.87 | 57.80 | 87.40 | 61.80 | 94.00 | 93.52 | 81.25 | |
| + Perm. | 59.33 | 76.99 | 1964.3 | 54.93 | 47.60 | 82.30 | 53.50 | 89.24 | 58.26 | 75.58 | ▼ 9.40 |
| | (▼ 1.33) | (▼ 0.27) | (▼ 18.2) | (▼ 4.93) | (▼ 10.20) | (▼ 5.10) | (▼ 8.30) | (▼ 4.76) | (▼ 35.26) | (▼ 5.67) | |
| QwenVL-2 | 50.67 | 77.06 | 2356.70 | 55.27 | 62.60 | 94.10 | 59.80 | 0.00 | 94.83 | 80.21 | |
| + Perm. | 48.67 | 77.19 | 2266.96 | 53.47 | 62.20 | 93.20 | 53.10 | 0.00 | 83.59 | 77.43 | ▼ 12.82 |
| | (▼ 2.00) | (▲ 0.13) | (▼ 89.74) | (▼ 1.80) | (▼ 0.40) | (▼ 0.90) | (▼ 6.70) | (▼ 0.00) | (▼ 11.25) | (▼ 2.78) | |
| Fuyu-8B | 30.00 | 40.33 | 0.00 | 19.67 | 16.30 | 0.00 | 0.00 | 0.00 | 15.81 | 0.00 | |
| + Perm. | 28.67 | 38.80 | 0.00 | 18.93 | 10.90 | 0.00 | 0.00 | 0.00 | 7.50 | 0.00 | ▼ 6.92 |
| | (▼ 1.33) | (▼ 1.54) | (▼ 0.00) | (▼ 0.73) | (▼ 5.40) | (▼ 44.00) | (▼ 0.00) | (▼ 0.00) | (▼ 8.31) | (▼ 0.00) | |

Table 4: Results of drop ratio ($\Delta$) when random permutation is applied. We run five experiments.

**Experimental Setup: Section 3.6.** We curated image classification datasets containing realistic and natural images across various domains. To explore the platonic representation hypothesis Huh et al. (2024), we first thoroughly examined its definition, as detailed in Appendix B. This process involved preparing a diverse set of LLMs, vision encoders, and vision encoders equipped with projectors in LVLMs. We also selected datasets for verifying cross-modal alignment, ensuring that they included natural and realistic images.

**Experimental Setup: Section 3.7.** We first clarified the definition of "importance score" and determined how to introduce noise into the visual patch tokens. This procedure is described in Appendix C.

# E  ADDITIONAL EXPERIMENTAL RESULTS

## E.1  PERMUTATION INVARIANCE.

As shown in Table 4, we investigate the extent to which other LVLMs exhibit permutation invariance under the same experimental settings described in Table 1. Overall, the Qwen2-VL-7B Wang et al. (2024) and Fuyu-8B models Bavishi et al. (2023) demonstrate permutation invariance on average, displaying patterns similar to those observed in the LLaVA-family models. A more detailed analysis across benchmarks reveals interesting patterns. In perception-focused benchmarks, such as MMVP, Q-Bench, MME, and MMStar (the latter two being integrated capability benchmarks that include perception-related tasks), the performance drop due to permutation is negligible. However, in text-rich benchmarks like MathVista and ChartQA, the performance drops significantly. These benchmarks require an understanding of detailed numerical information and highly structured geometric graphs, where maintaining the spatial structure of visual patch tokens is critical.

**Difficulty of Benchmark.** Interestingly, in the SQA$^I$ benchmark, which includes science-related datasets, and the AI2D benchmark, which consists of diagram images, the relatively small performance gap is noteworthy, even though these images are rich in detail. We speculate that this phenomenon might be influenced by the difficulty of the benchmark, particularly the "question type." Benchmarks typically include two question formats: (1) free-form and (2) multiple-choice questions (MCQ). We hypothesize that:

1. LLMs can often solve questions using their extensive commonsense reasoning, even without image perception. Li et al. (2024a); Fu et al. (2024)

2. MCQ formats may be easier for models compared to free-form questions due to the presence of preferred answer patterns or inherent biases in selection.

To investigate further, we conduct additional experiments comparing the difficulty of MathVista, ChartQA, SQA$^I$, and AI2D. We randomly select 500 samples from each dataset and, for MCQ

| Datasets | Question Type | Accuracy (%) | Don't Know (%) |
|---|---|---|---|
| MathVista | Free-Form | 0.3 | 82.1 |
| | MCQ | 36.8 | 0 |
| | Overall | 13.6 | 52.2 |
| ChartQA | Free-Form | 0 | 90 |
| SQA[I] | MCQ | 64.2 | 0 |
| AI2D | MCQ | 53.2 | 1.6 |

Table 5: Accuracy results of ChatGPT on four benchmarks for two different question types.

samples, include only those with four options. We then prompt ChatGPT (i.e., `gpt-3.5-turbo`) to answer these questions using the following templates:

---

**Prompt Template for MCQ**

Question: {question}
Choices:
{choices}
E: I don't know.

Please MUST generate only one option (A, B, C, D, E). Do not generate any explanation.
Answer:

---

**Prompt Template for Free-Form**

Question: {question}

Please provide your answer. If it is difficult to provide an answer, respond with "I don't know."

---

We added the "*I don't know*" option to prevent the model from guessing randomly. Table 5 show that ChatGPT performs better on MCQ-type benchmarks compared to free-form types. Moreover, ChatGPT achieves higher accuracy on AI2D and SQA[I] compared to MathVista and ChartQA. This supports the observation that LLLVMs exhibit less permutation invariance in these text-rich benchmarks, possibly due to the nature of the datasets and their question formats. For free-form questions, the "don't know" response rate is significantly higher, indicating that these benchmarks are more challenging. This highlights the need to minimize "blind" samples — questions solvable by LLMs without image perception — in benchmark design. Benchmarks should prioritize free-form questions to reduce potential selection bias Zheng et al. (2023a), as argued by recent studies Li et al. (2024a).

### E.2 SENSITIVITY TO SPATIAL STRUCTURES

As shown in Figure 11, we randomly shuffle image patches to evaluate their impact on model performance and observe that `Qwen2-VL` exhibits a similar tendency to LLaVA-family models. Specifically, we found that `Qwen2-VL` and `LLaVA-OneVision` are highly sensitive to spatial structures in text-rich benchmarks (e.g., MathVista, AI2D), which contain detailed numerical information. Notably, the performance of the `Qwen2-VL` model dropped significantly when the grid size was 2. To understand why `Qwen2-VL` is particularly sensitive, we hypothesize that this behavior is linked to its use of enhanced multi-modal rotary position embeddings (M-ROPE) Wang et al. (2024). This embedding mechanism likely contributes to the performance degradation observed when image patches are shuffled. Conversely, the model is relatively insensitive to spatial structures in perception-centric benchmarks (e.g., MMVP).

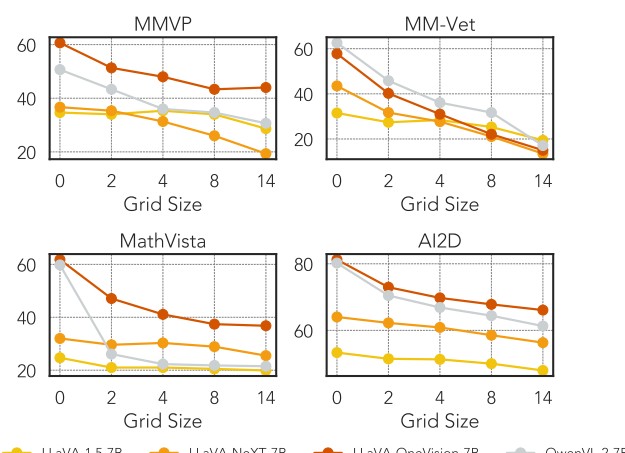

Figure 11: We present the performance across different grid sizes (2, 4, 8, 14) on the MMVP, MM-Vet, MathVista, and AI2D datasets, using four models: `LLaVA-1.5`, `LLaVA-NeXT`, `LLaVA-OneVision`, and `Qwen2-VL`.

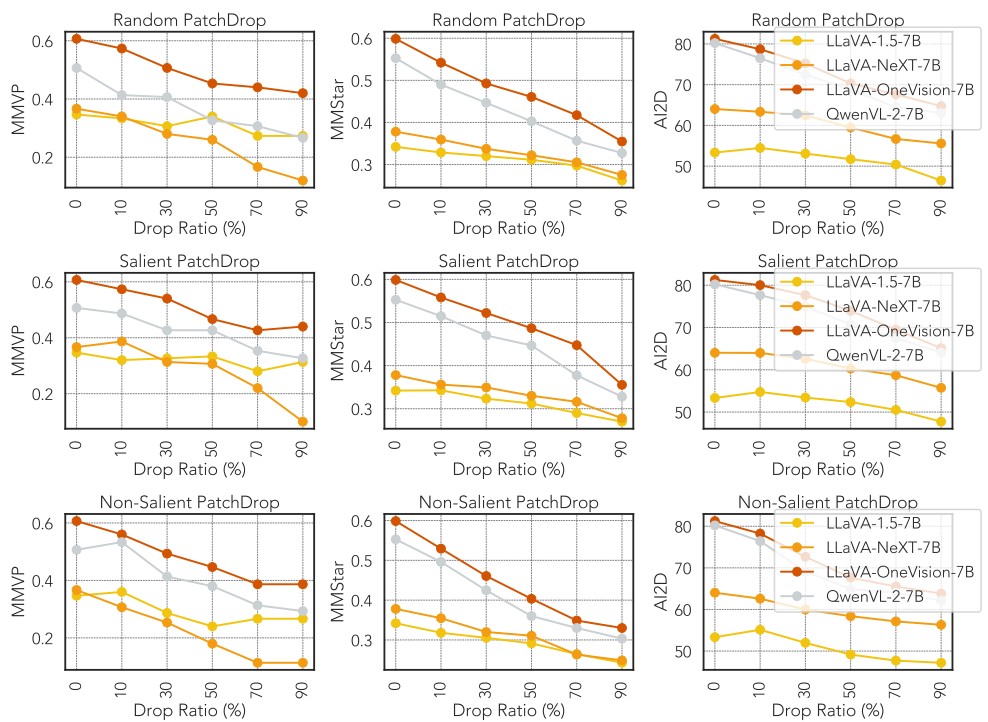

Figure 12: We present robustness performance under occlusion conditions.

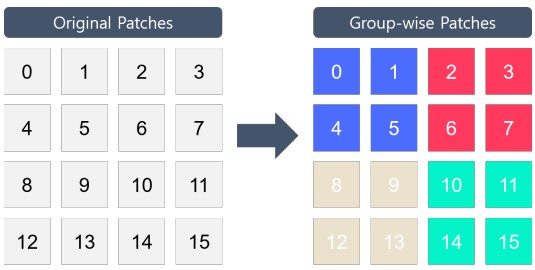

Figure 13: An illustration of group-wise patching.

### E.3 OCCLUSIONS

In Figure 12, we observe that the `Qwen2-VL` model exhibits a similar tendency to the LLaVA family models. Notably, the performance trend slope of the `Qwen2-VL` model closely resembles that of `LLaVA-OneVision`, suggesting that both models — currently high-performing LVLMs — share similar patterns. This alignment supports the generalizability of our observations. Specifically, LVLMs demonstrate relatively strong performance under occlusion. For instance, in the AI2D dataset, even when 50–70% of image patches are missing, the models can still provide correct answers to some extent. Moreover, in these scenarios, the `Qwen2-VL` and `LLaVA-OneVision` models outperform `LLaVA-1.5` and `LLaVA-NeXT`, even when no patches are missing. These results indicate that state-of-the-art LVLMs possess strong visual understanding capabilities. This suggests that improving visual understanding during training contributes significantly to high performance and robustness against occlusion.

### E.4 VARYING GRID SIZE FOR GROUPING STRATEGY

In Figure 1 and Figure 8, we group the nearest patches. For clarification, we visualize how the patches are grouped, as shown in Figure 13. Similar to the operation of a convolution layer, we group neighboring patches into a single group (indicated by the same color) and feed these groups into the model. Here, we vary the grid size, which corresponds to changing the number of elements in each group, and investigate whether the pattern observed in Figure 1 changes. We conduct additional experiments using a $3 \times 3$ grid of patches in Figure 14. We observe that increasing the number of grid patches leads to more precise observations. Compared to a $6 \times 6$ grid of patches, a $3 \times 3$ grid yields less precise observations. While conducting experiments on all visual patch tokens (576 for `LLaVA-1.5`) would provide the most precise interpretations, this approach is computationally intensive, as mentioned in Section 3.3. Therefore, we believe our chosen grid size strikes a reasonable balance for obtaining meaningful interpretations.

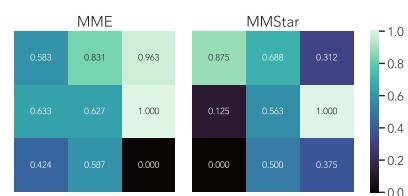

Figure 14: We demonstrate the extent to which group-wise visual tokens capture region-specific information (PIL) for `LLaVA-1.5-7B` on the MMStar (Chen et al., 2024a) and MME (Fu et al., 2023) when a $3 \times 3$ gird of patches. Darker regions indicate areas where the model retains more localized information for those specific groups.

### E.5 DETAILED ANALYSIS OF NUMERICAL INFORMATION

As shown in the above Table 6, in overall, the `Org.` ratio of LLVM generating "1" in free-form question types is reduced compared to the `Syn.` cases. This results suggest that LLVMs can effectively interpret and understand the detailed numerical information in the given image, thereby, the phenomenon that LLVM tend to use their commonsense reasoning is reduced. However, considering the ratio of `LLaVA-1.5` (44%), this ratio is not negligible. Therefore, in the future, we need to build more challenging benchmark that do not rely on the commonsense reasoning.

Additionally, we observe that most LVLMs prefer to answer "no" for `yes/no` question types in multiple-choice question (MCQ) formats. This suggests that, when presented with synthesized images, LVLMs struggle to solve the given questions effectively. Instead of attempting to provide an

| | Syn. | | | Orig. | | |
|---|---|---|---|---|---|---|
| Model | Freq. of 1 | No (%) | Precision | Recall | Freq. of 1 | Precision | Recall |
| LLaVA-1.5 | 81.0 | 64.0 | 49.2 | 36.8 | 44.4 | 59.4 | 43.7 |
| LLaVA-NeXT | 50.0 | 54.4 | 50.0 | 47.1 | 13.8 | 54.0 | 39.1 |
| Meteor | 9.5 | 82.8 | 55.2 | 18.4 | 7.5 | 78.0 | 36.8 |
| LLaVA-OneVision | 12.0 | 70.5 | 54.9 | 32.2 | 8.3 | 72.4 | 72.4 |

Table 6: Detailed analysis of the `Syn.` and `Orig.` versions of MathVista Lu et al. (2023). Precision and recall are reported for the `yes/no` question type.

| Datasets | Prompt Template for CLIP | Prompt Template for LLVM |
|---|---|---|
| Caltech101 | a photo of a {c}. | What is the object in the image? Please answer only a single object in {class_labels}. |
| CIFAR-100 | a photo of a {c}. | What is the object in the image? Please answer only a single object in {class_labels}. |
| Food101 | a photo of {c}, a type of food | What is the type of food in the image? Please answer only a single type of food in {class_labels}. |
| Pets | a photo of a {c}, a type of pet. | What is the type of pet in the image? Please answer only a single type of pet in {class_labels}. |
| Country211 | a photo showing the country of {c}. | What is the country in the image? Please answer only a single country in {class_labels}. |
| EuroSAT | a centered satellite photo of {c}. | What is the type of centered satellite in the image? Please answer only a single type of centered satellite in {class_labels}. |
| AirCraft | a photo of a {c}, a type of aircraft. | What is the type of aircraft in the image? Please answer only a single type of aircraft in {class_labels}. |

Table 7: Prompt templates used for evaluating CLIP and LLMs on zero-shot image classification tasks. The `c` represents a single class label, while `class_labels` refers to all class labels provided by each dataset.

answer based on the limited or unclear information available in the synthesized images, LVLMs tend to decline by answering "no," leading to an increased frequency of "no" responses compared to "yes." Furthermore, across all models, the `Org.` dataset consistently yields better performance in both precision and recall. This indicates that LVLMs face significant challenges in solving questions based on synthesized information. In the `Syn.` case, precision is consistently higher than recall, reflecting the tendency of LVLMs to output "no" answers more frequently than "yes" answers. This behavior underscores the challenges LVLMs face in effectively using synthesized visual information to provide accurate answers to `yes/no` questions.

### E.6 ADDITIONAL RESULTS OF CROSS-MODAL ALIGNMENT

**How to evaluate the zero-shot image classification task?** To evaluate CLIP models on the zero-shot classification task, we use the prompt templates provided by CLIP-Benchmark [7]. All the prompt templates we used are presented in Table 7. For evaluating LLVMs on the zero-shot image classification task, we design prompt templates inspired by those used for the CLIP model. Using these templates, the LLVM predicts a single class label. Based on the LLVM's generated answer, we then use ChatGPT to verify the prediction. Specifically, we utilize the following prompt: *Please only answer the question in yes or no. Is the "Prediction" correctly predicting the right 'Label'? Label: label; Prediction: outputs.* This evaluation method strictly follows the approach used in an existing study Zhai et al. (2024).

### E.7 ADDITIONAL RESULTS ON IMAGE CAPTIONING TASK

The evaluation benchmarks used in our experiments primarily consist of VQA tasks, which focus on binary, multiple-choice, and free-form question types. To address whether our claim regarding "permutation invariance" generalizes to other datasets, we conduct additional experiments using image captioning tasks. These tasks inherently require "visual processing capabilities," such as understanding attributes, viewpoints, scenes, and objects. For this investigation, we evaluate three standard datasets: COCO-Captions Lin et al. (2014) (Karpathy test set), NoCaps Agrawal et al. (2019) (validation set), and TextCaps Sidorov et al. (2020) (validation set). To generate captions, we followe the default prompting setup from `LMMs-Eval` [8], which uses the prompt: "*Please carefully observe the image and come up with a caption for the image.*" We employ standard evaluation metrics — ROUGE-L Lin (2004) and CIDEr Vedantam et al. (2015) — to assess performance.

---

[7] https://huggingface.co/clip-benchmark

[8] https://huggingface.co/lmms-lab

| | COCO-Captions | | NoCaps | | TextCaps | |
|---|---|---|---|---|---|---|
| LLVMs | ROUGE-L | CIDEr | ROUGE-L | CIDEr | ROUGE-L | CIDEr |
| LLaVA-1.5 | 22.01 | 0.97 | 25.34 | 1.52 | 22.46 | 6.09 |
| + Perm. | 22.62 | 1.26 | 26.05 | 2.89 | 22.94 | 7.28 |
| Avg. Δ | ▲ 0.62 | ▲ 0.29 | ▲ 0.71 | ▲ 1.37 | ▲ 0.48 | ▲ 1.19 |
| LLaVA-NeXT | 21.63 | 8.12 | 22.78 | 6.26 | 21.49 | 15.94 |
| + Perm. | 21.86 | 7.64 | 22.68 | 5.81 | 20.19 | 12.30 |
| Avg. Δ | ▲ 0.24 | ▼ 0.48 | ▼ 0.10 | ▼ 0.44 | ▼ 1.29 | ▼ 3.65 |
| LLaVA-OneVision | 57.23 | 116.25 | 56.09 | 86.60 | 44.58 | 72.69 |
| + Perm. | 56.70 | 116.17 | 56.36 | 85.94 | 44.19 | 68.18 |
| Avg. Δ | ▼ 0.53 | ▼ 0.08 | ▲ 0.26 | ▼ 0.66 | ▼ 0.39 | ▼ 4.52 |
| Qwen2-VL | 39.98 | 44.61 | 44.01 | 39.37 | 35.80 | 46.86 |
| + Perm. | 37.19 | 39.29 | 42.70 | 38.35 | 35.31 | 44.64 |
| Avg. Δ | ▼ 2.79 | ▼ 5.33 | ▼ 1.31 | ▼ 1.02 | ▼ 0.49 | ▼ 2.22 |

Table 8: Results of drop ratio (Δ) when random permutation is applied. We run five experiments.

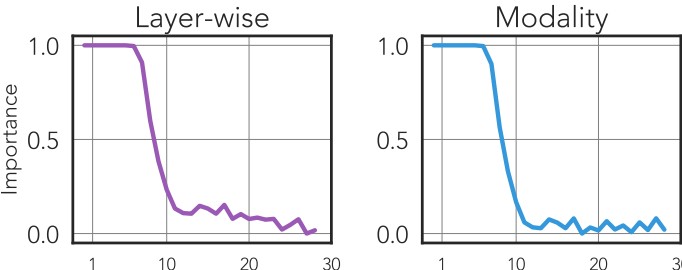

Figure 15: We present the results of (left) layer-wise importance and (right) modality importance within the layers on MME Fu et al. (2023) dataset.

As shown in Table 8, we observe similar trends across image captioning datasets: **most LLMs exhibit permutation invariance.** Interestingly, on the TextCaps dataset, the performance drop is more pronounced compared to other datasets, suggesting relatively greater permutation variance. TextCaps contains more complex images (e.g., those with detailed numerical information) compared to the other datasets, which may explain this phenomenon. When comparing these findings to those in Table 1, we note that in perception-related tasks (e.g., involving natural scenes), LLVMs generally exhibit permutation invariance. However, in reasoning-related tasks (e.g., MathVista) involving images with complex structures (e.g., charts or diagrams), LLMs demonstrate greater permutation variance. This suggests that maintaining the geometric or positional structure of plots and charts is crucial.

### E.8 ADDITIONAL RESULTS ON LAYER & MODALITY IMPORTANCE

Figure 15 (left) shows that the lower layers (< 10) play a crucial role in handling integrated capabilities. Meanwhile, Figure 15 (right) demonstrates that in the lower layers (< 12), the image modality is more important than the text modality. Overall, the tendencies observed on the MME dataset are similar to those on the MMStar dataset, as shown in Figure 9. However, a key difference lies in the layer index at which the modality importance shifts; for the MME dataset, this transition occurs at a higher layer index. Based on these results, we hypothesize that LLVMs allocate more effort to understanding the given images on the MME dataset compared to the MMStar dataset. One of the possible reason is that the images in the MME dataset are more challenging for the model

to comprehend, but we can not guarantee this reason is correct, therefore, Further investigation is required to validate this assumption in future studies.

## F ADDITIONAL RELATED WORKS

**Model-Stitching.** The model-stitching (Lenc & Vedaldi, 2015; Bansal et al., 2021) is a technique first introduced to study the internal representations of neural networks by measuring the representational similarity between two given models. Consider two models defined as $f = f^m \circ \cdots \circ f^1$ and $g = g^n \circ \cdots \circ g^1$. Specifically, the *stitched* model is formalized as $\mathcal{F} = g^n \circ \cdots \circ g^{k+1} \circ s \circ f^k \circ \cdots \circ f^1$, where $s$ is a simple stitching layer (e.g., a linear layer or a $1 \times 1$ convolution). Therefore, even if the two models $f$ and $g$ differ in training methodology (e.g., supervised vs. self-supervised) or modalities (e.g., text vs. image), if $\mathcal{F}$ exhibits good performance, then $f$ and $g$ have strongly correlated and compatible internal representations at layer $k$, apart from the stitching layer $s$. Merullo et al. (2022) have the similar concept of *model-stitching* to verify their strong hypothesis that the conceptual representations from a frozen LLM and a visual encoder are sufficiently similar such that a simple linear mapping layer can align them.

## G ADDITIONAL EXAMPLES OF SYNTHESIZED IMAGES

We provide additional examples of synthesized images in Figure 16.

## H ADDITIONAL EXAMPLES OF SHUFFLED IMAGES

We provide additional examples of shuffled images in Figure 17.

## I ADDITIONAL EXAMPLES OF OCCLUDED IMAGES

We provide additional examples of occluded images in Figure 18.

## J DESCRIPTION OF EVALUATION BENCHMARKS

- **MM-Vet** (Yu et al., 2023) dataset is a benchmark designed to evaluate large vision-language models (LVLMs) across six core vision-language (VL) capabilities: recognition, knowledge, optical character recognition (OCR), spatial awareness, language generation, and mathematical reasoning. The dataset includes open-ended, real-world questions based on image-text pairs, requiring models to integrate multiple capabilities to solve complex tasks. MM-Vet benchmark consists of 200 images paired with 218 open-ended questions.

- **Q-Bench** (Wu et al., 2023) evaluates the capabilities of large vision-language models in three main areas related to low-level vision tasks. These tasks focus on evaluating how well LVLMs can perform basic low-level perception tasks that are traditionally associated with human visual perception. In the Q-Bench dataset, the questions are of three types: Yes-or-No, What, and How.

  - **Low-Level Visual Perception**: Assesses how accurately LVLMs can answer questions about low-level image attributes (e.g., clarity, color, distortion). LLVisionQA dataset includes 2,990 images, each with a corresponding question about low-level features.
  - **Low-Level Visual Description**: Evaluates the ability of LVLMs to describe images. LLDescribe dataset has 499 images with expert-labeled descriptions averaging 58 words each. LVLMs are compared against these to assess completeness, preciseness, and relevance.
  - **Visual Quality Assessment**: Evaluates LVLMs' ability to predict quantifiable quality scores for images by assessing how well they align with human-rated mean opinion scores (MOS) on low-level visual appearances, using 81,284 samples.

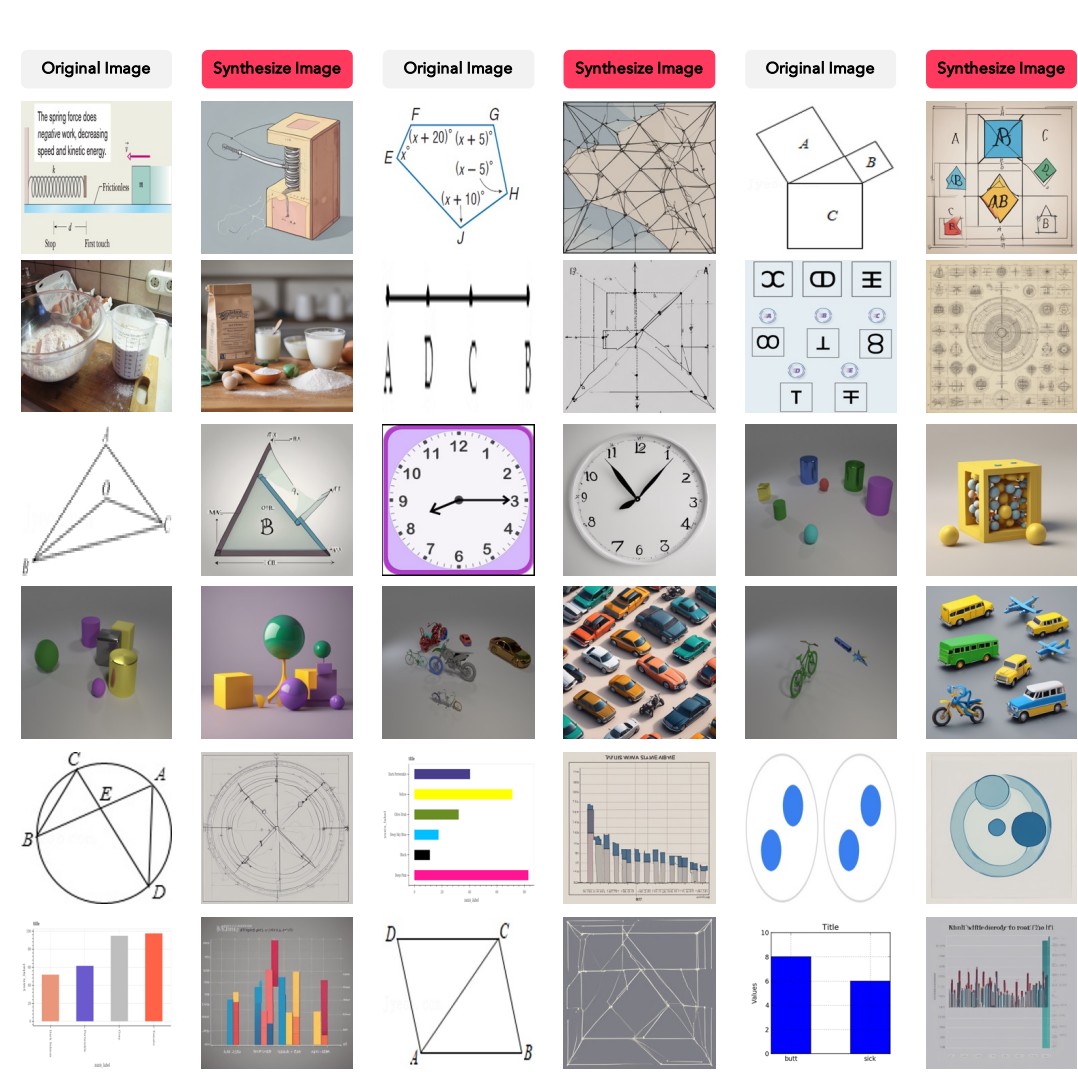

Figure 16: Examples of synthesized images from MathVista Lu et al. (2023).

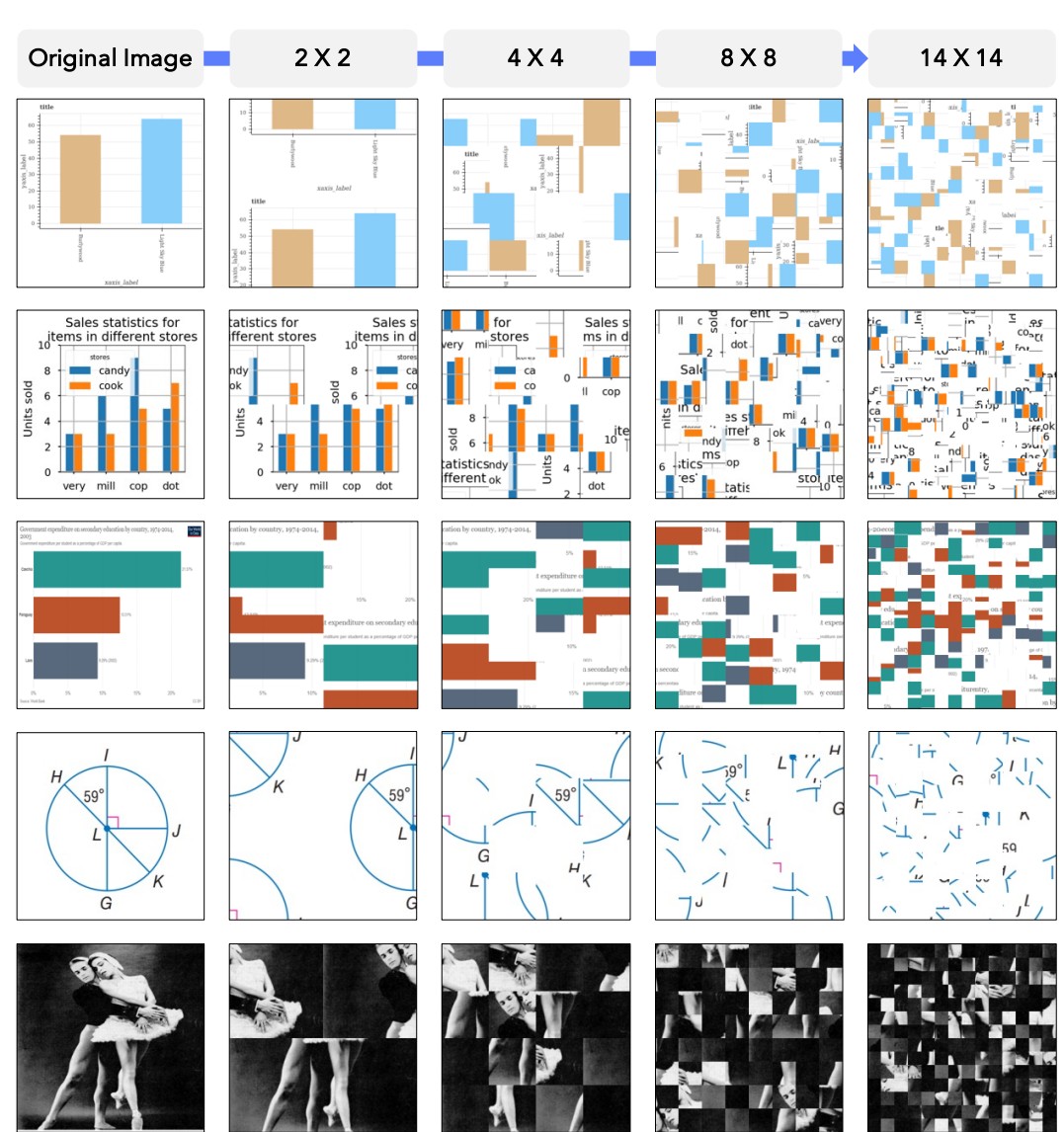

Figure 17: Examples of synthesized images from MM-Vet Yu et al. (2023).

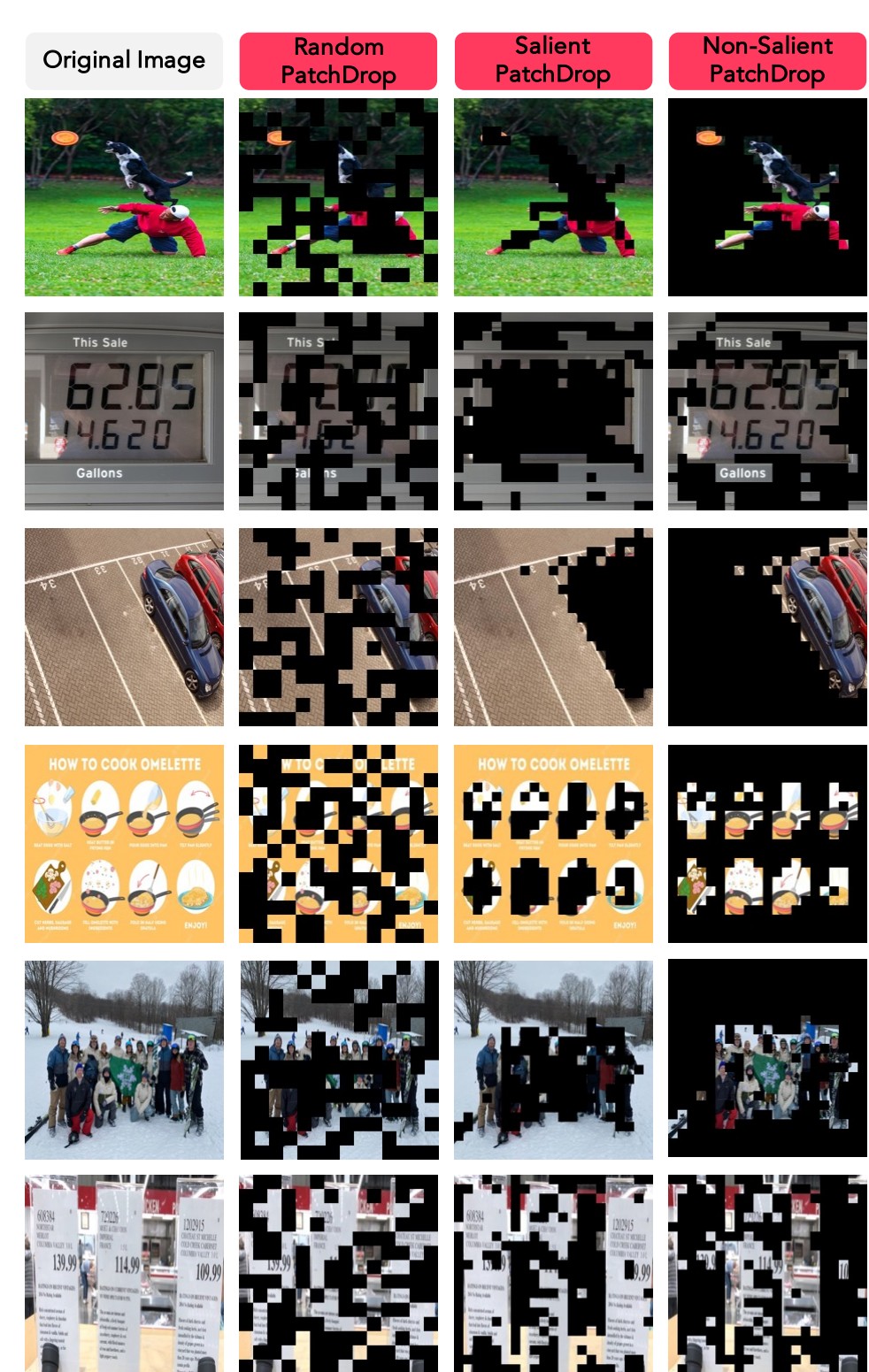

Figure 18: Examples of occluded images from MME Fu et al. (2023).

- **SQA-IMG** (Lu et al., 2022a) is a portion of the Science Question Answering (SQA) dataset that contains questions from a wide range of scientific domains, each paired with corresponding image contexts. The dataset includes 10,332 examples of multimodal multiple-choice questions, along with lectures and explanations that detail the reasoning behind the correct answers.

- **ChartQA** (Masry et al., 2022) dataset is a benchmark designed to test AI models on their ability to perform question-answering tasks over various types of charts. It focuses specifically on questions requiring complex reasoning, such as visual and logical interpretation, going beyond simpler template-based datasets. ChartQA includes 9,608 human-authored open-ended questions as well as 23,111 questions that are automatically generated from chart summaries.

- **SEED-IMG** (Li et al., 2023), a subset of SEED-Bench, focuses on evaluating spatial comprehension of images by testing models on various dimensions like scene understanding, object identification, and spatial relationships. In terms of scale, the dataset includes 19,000 multiple-choice questions that evaluate both image and video comprehension, covering 12 evaluation dimensions such as scene understanding, instance identity, spatial relations, and action recognition.

- **MME** (Fu et al., 2023) evaluates both perception and cognition abilities of LVLMs. It features 14 subtasks, including recognition tasks (such as object existence, count, position, color) and reasoning tasks (such as commonsense reasoning, numerical calculation, text translation, and code reasoning). MME uses manually created instruction-answer pairs, ensuring no overlap with public datasets. MME uses "yes/no" responses for quantitative evaluations.

- **MathVista** (Lu et al., 2023) is a benchmark designed to evaluate the mathematical reasoning capabilities of foundation models in visual contexts. It integrates challenges from diverse mathematical and visual tasks, with a focus on fine-grained, deep visual understanding and compositional reasoning. MathVista consists of 6,141 examples including 3,392 multiple-choice questions and 2,749 free-form questions derived from 28 existing multimodal datasets and 3 newly created datasets: IQTest, FunctionQA, and PaperQA.

- **LLaVA-W** (Liu et al., 2024c) is a challenging evaluation benchmark created to assess the generalization and instruction-following capabilities LVLMs in complex, real-world situations. It consists of 24 images and 60 questions, including diverse scenes like indoor environments, outdoor settings, memes, paintings, and sketches. Each image is associated with a highly detailed and manually curated description, and the questions focus on extracting intricate details and reasoning about the visual content. LLaVA-W involves a variety of tasks, including detailed descriptions, conversational answers, and complex reasoning.

- **MMStar** (Chen et al., 2024a) is a vision-dependent multimodal benchmark designed to evaluate the multimodal capabilities of LVLMs. It addresses two main issues identified in previous benchmarks: the reliance on textual information without visual input and data leakage during training. MMStar is composed of 1,500 samples carefully selected to ensure that visual content is necessary for solving each problem. MMStar evaluates six core capabilities across 18 detailed axes, which include tasks like image perception and logical reasoning. MMStar uses multiple-choice as the primary answer type.

- **MMVP** (Tong et al., 2024) evaluates the visual grounding capabilities of large vision-language models by identifying scenarios where they fail to distinguish simple visual patterns in images. These patterns include aspects like orientation, counting, viewpoint, and relational context. The benchmark is constructed using 150 pairs of images, resulting in 300 multiple-choice questions.

## K  DESCRIPTION OF EVALUATION LVLMS

- **LLaVA-1.5** (Liu et al., 2024a) incorporates academic task-oriented datasets to enhance performance in VQA tasks and features an MLP vision-language connector, which improves upon the original linear layer utilized in LLaVA (Liu et al., 2024c). It uses CLIP ViT-L/14 (Radford et al., 2021) with a 336px resolution as its vision encoder, resulting in a total of $(336/14)^2 = 576$ visual tokens. LLaVA-1.5 is built on Vicuna with either 7B or

13B parameters. The training dataset includes 558K samples for pre-training and 665K for fine-tuning, totaling 1.2M image-text pairs from publicly available datasets

- **LLaVA-NeXT** (Liu et al., 2024b) (also known as LLaVA-1.6) enhances visual reasoning, OCR, and world knowledge, offering four times higher image resolution (up to 1344x336) and improved performance in visual conversations. Its architecture includes a CLIP ViT-L/14 as a vision encoder, paired with Vicuna models ranging from 7B to 34B as a backbone language model. It utilizes 1.3M visual instruction tuning data samples for training, maintaining efficiency with approximately one day of training on 32 A100 GPUs. The architecture's high resolution and dynamic grid scheme improve detailed image processing capabilities.

- **LLaVA-OneVision** (Li et al., 2024c) is a LVLM designed for task transfer across single-image, multi-image, and video scenarios, with strong capabilities in video understanding through image-to-video task transfer. Its architecture consists of a Qwen2 language model (Yang et al., 2024) with 8B to 72B parameters, and the SigLIP vision encoder (Zhai et al., 2023), which processes images at a base resolution of 384x384, producing 729 visual tokens. The model employs a 2-layer MLP projector. The training utilized 3.2M single-image data samples and 1.6M multi-modal data samples, focusing on high-quality visual instruction tuning data to enhance its multimodal capabilities.

- **Meteor** (Lee et al., 2024c) is a large vision-language model that uniquely embeds multi-faceted rationales using a Mamba-based architecture (Gu & Dao, 2023), enabling efficient processing of lengthy rationales to enhance its vision-language understanding. This approach allows Meteor to achieve superior performance without scaling up model size or employing additional vision encoders. Its architecture includes a CLIP-L/14 vision encoder with an image resolution of 490x490, comprising 428M parameters, and InternLM2-7B (Cai et al., 2024) as a foundational LLM. Meteor was trained on 2.1M question-answer pairs, with 1.1M curated triples.

- **TroL** (Lee et al., 2024b) uses a unique characteristic called layer traversing, which reuses layers in a token-wise manner, allowing it to simulate retracing the answering process without physically adding more layers, making it efficient despite smaller model sizes. TroL uses CLIP-L and InternViT as vision encoders, containing 428M and 300M parameters, respectively, and supports 24 layers. The image resolution is adjusted using MLPs in the vision projector. For its foundational LLM, TroL utilizes Phi-3-mini with 3.8B parameters and InternLM2 with 1.8B and 7B parameters. The training dataset comprises 2.3M visual instruction tuning samples.

