# OpenReview forum: "Intriguing Properties of Large Language and Vision Models"
_ICLR.cc/2025/Conference — Submitted to ICLR 2025_

### Official Review · Reviewer_Tcxa · 2024-11-02

**Soundness:** 3
**Presentation:** 3
**Contribution:** 3
**Rating:** 6
**Confidence:** 4

**Summary:**

This paper presents a detailed investigation of how vision is perceived in large language and vision models. Comprehensive experiments reveal the properties of LLVMs from multiple aspects including permutation invariance, robustness, math reasoning, alignment preserving and importance.

**Strengths:**

This paper elaborates on a critical problem in multimodal LLMs that how vision is perceived. It is a profound question which could benefit the future development of MLLMs.

**Weaknesses:**

The examples to be investigated in this paper are mainly from the LLaVA family whose vision representation is continuous. An analysis on discrete vision representation would be interesting.

**Questions:**

Please see weakness.

---

> ### Author Response · Authors · 2024-11-22
> **Response by Authors**
>
> We greatly appreciate the positive feedback recognizing that our work addresses important questions and highlights intriguing properties.
>
> ---
>
> **Q1.** The examples to be investigated in this paper are mainly from the LLaVA family whose vision representation is continuous. An analysis on discrete vision representation would be interesting.
>
> **A1.** Thank you for your insightful comment. We agree that demonstrating **"model-side generalizability"** is crucial to solidify and validate our findings. In particular, evaluating LLVM models that utilize **"discrete vision representations"** would indeed enrich our analysis and improve robustness. However, we want to ensure that we have accurately interpreted the reviewer's intention regarding "discrete vision representation."
>
> Based on our understanding, "discrete vision representation" may refer to **LLVM models that do not rely on a dedicated vision encoder**. Instead, these models process **raw image patches (in pixels)** directly, projecting them through a simple linear layer into the transformer backbone. Such approaches have been explored in recent works [1-2] for their efficiency, flexibility in image preprocessing, and support for arbitrary image resolutions.
>
> To address this point, we conducted additional experiments using the **Fuyu-8B model** [1], a decoder-only transformer architecture that does not employ a specialized vision encoder but instead projects image patches linearly into the first transformer layer.
>
> We evaluated the **permutation invariance** of the Fuyu-8B model across the MMVP, Q-Bench, MMStar, MMVet, and ChartQA datasets, following the same settings as in Table 1 of the main paper. The results are summarized below:
>
> | Model | MMVP | Q-Bench | MMStar | MM-Vet | ChartQA | Avg. |
> | --- | --- | --- | --- | --- | --- | --- |
> | **Fuyu-8B** | 30.00 | 40.33 | 19.67 | 16.30 | 15.81 | 24.42 |
> | **+ Perm.** | 28.67 | 38.80 | 18.93 | 10.90 | 7.50 | 20.96 |
> | **Δ** | -1.33 | -1.54 | -0.73 | -5.40 | -8.31 | -3.46 |
>
> As shown in the table, the Fuyu-8B model exhibits a similar trend to the LLaVA-series models: the performance drop under permutation invariance is generally modest (average drop: 3.46). In more detail:
>
> - For **perception-related tasks** (e.g., MMVP, Q-Bench), the model demonstrates permutation invariance.
> - For **reasoning-related tasks** (e.g., ChartQA), which require detailed numerical understanding and structured geometric reasoning, the model shows permutation variance. These results align with the patterns observed in Table 1 of our main paper.
> - For **MMStar and MM-Vet tasks**, which are integrated capability benchmarks involving both perception and reasoning tasks, the model demonstrates *permutation invariance* in MMStar but *permutation variance* in MM-Vet. Surprisingly, despite both benchmarks being designed to evaluate integrated capabilities, the model's behavior with respect to permutation differs significantly. From these observations, we hypothesize that this discrepancy may be attributed to the *question type*: MMStar primarily consists of multiple-choice questions (MCQs), while MM-Vet requires free-form answers. Consequently, we posit that LLVMs can still choose the correct option from four choices in MMStar even when the visual patch tokens are randomly permuted. In contrast, MM-Vet's requirement for generating free-form answers leads to a relatively larger performance drop when visual patch tokens are permuted. Based on these findings, while MMStar highlights "blind" pairs (i.e., cases where the LLM can answer the question using commonsense reasoning without perceiving the image) in their paper, we argue that there remains substantial room for improvement. Specifically, there is a need for more challenging free-form benchmarks, such as multi-turn conversational tasks, to better evaluate LLVM capabilities.
>
> This suggests that reasoning-intensive benchmarks depend on preserving the spatial structure of visual patch tokens, highlighting the importance of spatial integrity for tasks requiring detailed numerical or geometric comprehension.
>
> While this experiment demonstrates "model-side generalizability" to discrete vision representations, we acknowledge that additional investigations would further strengthen our analysis. However, due to time constraints and some ambiguity in the reviewer's definition of "discrete vision representations," we focused on permutation invariance for now. After clarifying the reviewer's intent, we will conduct additional experiments and continuously update our findings.
>
> We would like to kindly ask if our understanding of the "discrete vision representation" in LLVM, as mentioned by the reviewer, aligns with your intention. If not, could you please provide a few example models for clarification?
>
> ---
>
> **References**
>
> [1] https://www.adept.ai/blog/fuyu-8b
>
> [2] Chen, Yangyi, et al. "A Single Transformer for Scalable Vision-Language Modeling." *arXiv preprint arXiv:2407.06438* (2024).

---

> > ### Comment · Reviewer_Tcxa · 2024-12-02
> >
> > I appreciate the detailed response from the authors. My question has been well addressed. After checking comments from other reviewers, I would like to maintain my rating for a weak acceptance.

---

> > > ### Author Response · Authors · 2024-12-02
> > > **Response by Authors**
> > >
> > > We are glad to hear that our response addresses your concern effectively. Thank you for providing helpful comments; we will incorporate these results into our revised paper.

---

### Official Review · Reviewer_ePc1 · 2024-11-03

**Soundness:** 2
**Presentation:** 3
**Contribution:** 2
**Rating:** 6
**Confidence:** 3

**Summary:**

This paper presents an empirical study of large vision language models with a focus on the llava model families using perception and reasoning tasks across various benchmarks. The authors revealed several findings of LVMs including image patch permutation invariance, overfitted cross-modal alignment, importance of lower block layers, etc..

**Strengths:**

The paper comprehensively evaluated several VLMs across various perception and reasoning benchmark to better understand their behavior and performance. These empirical findings could be valuable providing insights for model developments.

**Weaknesses:**

1. While the paper shows some good empirical studies of the LVM performance, they are loosely connected to the suggested future directions. This makes the contribution of the paper less clear. For example, the authors suggested "deeply consider innovative model architectures" in Sec 4 for enhancing cross modal alignment, yet it's not clear how this related to the empirical findings discussed in the previous sections and what necessary enhancements are entailed from their analysis.

2. The authors should consider toning down the rhetoric of the study from general LVMs to llava model families. Reported results do not show whether these findings would generalize to other LVMs as they are not evaluated yet.

3. Some of the findings are more or less expected, e.g., some degree of robustness to occlusion due to the robustness from image encoder. The conclusion from these findings are not clear. I suggest the authors append more focused discussions on a few key findings and tie them to any future directions or proposed improvements.

**Questions:**

Please see weaknesses above.

---

> ### Author Response · Authors · 2024-11-24
> **Response by Authors (1/3)**
>
> We appreciate the positive feedbacks regarding our comprehensive evaluation and valuable of our intriguing findings.
>
> ---
>
> **Q1.** While the paper shows some good empirical studies of the LVM performance, they are loosely connected to the suggested future directions. This makes the contribution of the paper less clear. For example, the authors suggested "deeply consider innovative model architectures" in Sec 4 for enhancing cross modal alignment, yet it's not clear how this related to the empirical findings discussed in the previous sections and what necessary enhancements are entailed from their analysis.
>
> **A1.** Thank you for this insightful feedback about strengthening the connection between our empirical findings and proposed future directions. For our detailed response, please refer to the "Common Response" post.
>
> ---
>
> **Q3.** Some of the findings are more or less expected, e.g., some degree of robustness to occlusion due to the robustness from image encoder. The conclusion from these findings are not clear. I suggest the authors append more focused discussions on a few key findings and tie them to any future directions or proposed improvements.
>
> **A3.** Thank you for this thoughtful feedback. To enhance the value of our findings, we have added specific future directions for each observation (detailed in Appendix A). For our complete response, please refer to the "Common Response" post.

---

> ### Author Response · Authors · 2024-11-24
> **Response by Authors (2/3)**
>
> **Q2.** The authors should consider toning down the rhetoric of the study from general LVMs to llava model families. Reported results do not show whether these findings would generalize to other LVMs as they are not evaluated yet.
>
> **A2.** Thanks for the thorough feedback regarding “model-side generalizability”. To address this concern, we conducted additional experiments on **Qwen2-VL**, a leading LVLM known for its robust performance and advanced features. Specifically, we investigated key aspects such as **permutation invariance**, **sensitivity to spatial structure**, and **occlusion** to demonstrate the generalizability of our observations beyond the LLaVA family.
>
> **Additional Results: Permutation Invariance (Table 4 in Appendix E.1.)**
>
> | LLVMs | MMVP | Q-Bench | MME | MMStar | MM-Vet | LLaVA$^W$ | MathVista | SQA$^I$ | ChartQA | AI2D | Avg. Δ |
> | --- | --- | --- | --- | --- | --- | --- | --- | --- | --- | --- | --- |
> | LLaVA-1.5 | 34.67 | 59.73 | 1850.07 | 34.20 | 31.50 | 67.50 | 24.70 | 65.59 | 16.92 | 53.34 | - |
> | + Perm. | 36.00 (🔼 1.33) | 59.60 (🔻 0.13) | 1874.60 (🔼 24.53) | 33.33 (🔻 0.87) | 30.40 (🔻 1.10) | 66.20 (🔻 1.30) | 21.20 (🔻 3.50) | 65.44 (🔻 0.15) | 14.08 (🔻 2.84) | 52.69 (🔻 0.65) | 🔻 0.59 |
> | LLaVA-NeXT | 36.67 | 63.55 | 1874.42 | 37.80 | 43.50 | 75.50 | 32.00 | 62.12 | 66.06 | 64.02 | - |
> | + Perm. | 37.33 (🔼 0.67) | 62.54 (🔻 1.00) | 1890.19 (🔼 15.78) | 36.87 (🔻 0.93) | 43.40 (🔻 0.10) | 75.80 (🔻 0.30) | 21.70 (🔻 10.30) | 62.12 (🔻 0.00) | 34.55 (🔻 31.51) | 64.02 (🔻 0.00) | 🔻 2.71 |
> | LLaVA-OneVision | 60.67 | 77.26 | 1982.50 | 59.87 | 57.80 | 87.40 | 61.80 | 94.00 | 93.52 | 81.25 | - |
> | + Perm. | 59.33 (🔻 1.33) | 76.99 (🔻 0.27) | 1964.30 (🔻 18.20) | 54.93 (🔻 4.93) | 47.60 (🔻 10.20) | 82.30 (🔻 5.10) | 53.50 (🔻 8.30) | 89.24 (🔻 4.76) | 58.26 (🔻 35.26) | 75.58 (🔻 5.67) | 🔻 9.40 |
> | Qwen2-VL-7B | 50.67 | 77.06 | 2356.70 | 55.27 | 62.60 | 94.10 | 59.80 | 0.00 | 94.83 | 80.21 | - |
> | + Perm. | 48.67(🔻 2.00) | 77.19 (🔼 0.13) | 2266.96 (🔻 89.74) | 53.47 (🔻 1.80) | 62.20 (🔻 0.40) | 93.20 (🔻 0.90) | 53.10 (🔻 6.70) | 0.00 (🔻 0.00) | 83.59 (🔻 11.25) | 77.43 (🔻 2.78) | 🔻 12.82 |
> - *Note: Due to the large number of samples in the SQA-IMG dataset, we have not yet conducted the corresponding experiment for Qwen2-VL-7B, but we will include it in our paper.*
>
> We investigate the extent to which other LVLMs exhibit permutation invariance under the same experimental settings described in Table 1. Overall, the Qwen2-VL-7B and Fuyu-8B models demonstrate permutation invariance on average, displaying patterns similar to those observed in the LLaVA-family models. A more detailed analysis across benchmarks reveals interesting patterns. In perception-focused benchmarks, such as MMVP, Q-Bench, MME, and MMStar (the latter two being integrated capability benchmarks that include perception-related tasks), the performance drop due to permutation is negligible. However, in text-rich benchmarks like MathVista and ChartQA, the performance drops significantly. These benchmarks require an understanding of detailed numerical information and highly structured geometric graphs, where maintaining the spatial structure of visual patch tokens is critical.
>
> **Additional Results: Sensitivity to Spatial Structures (Figure 11 in Appendix E.2.)**
>
> We randomly shuffled image patches to evaluate their impact on model performance and observed that Qwen2-VL-7B exhibits a similar tendency to LLaVA-family models. Specifically, we found that Qwen2-VL and LLaVA-OneVision are highly sensitive to spatial structures in text-rich benchmarks (e.g., MathVista, AI2D), which contain detailed numerical information. Notably, the performance of the Qwen2-VL model dropped significantly when the grid size is 2. To understand why Qwen2-VL is particularly sensitive, we hypothesize that this behavior is linked to its use of enhanced multi-modal rotary position embeddings (M-ROPE). This embedding mechanism likely contributes to the performance degradation observed when image patches are shuffled. Conversely, the model is relatively insensitive to spatial structures in perception-centric benchmarks (e.g., MMVP). We speculate that this insensitivity arises from the robustness of vision encoders, as reported in a prior study, which suggest that ViT-based models are  robust to spatial disruptions in datasets like ImageNet. However, text-rich images, unlike perception-related images, include intricate numerical details. Shuffling image patches in such cases disrupts geometric relationships or relative magnitudes in plots or charts, making accurate interpretation of these images significantly more challenging.

---

> ### Author Response · Authors · 2024-11-24
> **Response by Authors (3/3)**
>
> **Additional Results: Occlusions (Figure 12 in Appendix E.3.)**
>
> In Figure 12, we observe that the Qwen2-VL-7B model exhibits a similar tendency to the LLaVA family models. Notably, the performance trend slope of the Qwen2-VL-7B model closely resembles that of LLaVA-OneVision, suggesting that both models — currently high-performing LVLMs — share similar patterns. This alignment supports the generalizability of our observations. Specifically, LVLMs demonstrate relatively strong performance under occlusion. For instance, in the AI2D dataset, even when 50–70% of image patches are missing, the models can still provide correct answers to some extent. Moreover, in these scenarios, the Qwen2-VL and LLaVA-OneVision models outperform LLaVA-1.5 and LLaVA-NeXT, even when no patches are missing. These results indicate that state-of-the-art LVLMs possess strong visual understanding capabilities. This suggests that improving visual understanding during training contributes significantly to high performance and robustness against occlusion.
>
> Here, we aim to demonstrate “model-side generalizability” in terms of permutation invariance, sensitivity to spatial structure, and occlusion. Overall, the Qwen2-VL model exhibits similar patterns across all three experiments, suggesting that our findings generalize to other LVLMs. However, as mentioned earlier, there is a wide variety of LVLMs in this field. To address this, we plan to continuously update our work by conducting additional experiments on other LVLMs to further evaluate the generalizability of our findings. We appreciate the reviewer’s comment and are committed to addressing it.

---

> > ### Author Response · Authors · 2024-11-26
> > **Response by Authors**
> >
> > We are very pleased that our rebuttals effectively addressed your concerns, resulting in an increase in the original score from 5 to 6. We sincerely appreciate the reviewer’s thoughtful feedback.

---

### Official Review · Reviewer_v3XH · 2024-11-03

**Soundness:** 2
**Presentation:** 3
**Contribution:** 2
**Rating:** 5
**Confidence:** 3

**Summary:**

The paper analyzes some properties of LLVMs with respect to permutation invariance, mathematical reasoning, robustness to perturbations, and other aspects. The problem tackled is interesting and relevant to the VLM community. The authors analyze a wide variety of VLMs properties by focusing on the LLAVA family and a diverse set of classification/instruction following benchmarks.

**Strengths:**

* The paper addresses an important and timely problem that is of interest to the VLM community.
* Some observations made by the authors are insightful and have the potential to contribute to the understanding of VLMs.
* The set of analysis is diverse and has the potential to inform the  development of a new set of architectures/benchmark.
* Detailed set of benchmark datasets in the Appendix.

**Weaknesses:**

* Many conclusions are not clearly backed up by evidence or rigorous analysis, which undermines their validity (I've given some examples in the questions section)
* Some observations appear to be cherry-picked (by dataset or sample?), which raises concerns about the representativeness and generalizability of the results. For instance, the authors want to evaluate the LLaVA family of models but some results are given on a single model and data sample while the conclusion is general (Figure 1 and Section 3.7 for instance, i've given more specific questions in the next section)
* The choice of datasets, metrics, and evaluation methods for different parts of the analysis is often unclear or unjustified, making it difficult to assess the generalizability of the findings.

**Questions:**

* The article focus on the LLaVA family. Could the authors  explain their rationale for focusing on the LLaVA family and discuss potential limitations in generalizing these results to other LVLM architectures?
* In Section 3.3, how is the grouping done to quantify local information in patches? How are neighbours defined, and what is the impact of this definition on the results?
* The explanation of backpropagation and loss signal coming from the assistant in line 234 is unclear. Can the authors provide more details or clarify the average performance drop, which seems to be benchmark-dependent?
* In Table 2, what setting is used to measure the frequency of outputting 1 (synthetic or original)? Is there a difference between these cases? Can the authors also report the frequency of outputting "no" and precision/recall for yes/no questions?
* Evaluating the loss of  cross-modal alignment is interesting. However the evaluation method used in Table 3 is unclear and may not be fair compared to a contrastive approach. Could the authors give more details about their evaluation setup (prompt used?) as it is known that prompt optimization impacts even zero-shot CLIP models classification performance.  Can the authors consider alternative approaches, such as log-likelihood comparison of description prompts (e.g., "Describe the image: This is an image of a [class label]")? As it is done in captioning models evaluation for instance?
* How is the 1k subset obtained in Table 3?
* The section on shared world concepts is hard to follow. What metric is used to measure alignment, and why was the DOCCI dataset chosen? How is alignment measured with long captions given CLIP's 77-tokens context?
* Are the local scores in Figure 8 averaged over the whole dataset or just one example? Same question for Figure 1 ?
* Can the authors comment on the choices of datasets for different parts of the analysis (e.g., MMVet for localization of information, MMStar for layers importance)? Are these results model-, dataset-, or sample-specific, and how do they impact the generalizability of the findings? I know this question is quite general but the justification is not clear to me and the drawn conclusions seem to be a general so I would except such visualizations to be performed on a subset of the different datasets.
* The conclusion drawn from Figure 9 lacks a clear explanation of how the importance score is obtained. Can the authors provide more details on the method used (e.g., noised input or weights, which proportion)?

* In general it would help if the authors could :
1. Provide explicit justifications for their dataset choices in each analysis section
2. Discuss potential limitations in generalizing results from specific datasets to broader conclusions
3. Consider extending their analysis to multiple datasets where feasible, or explain why this might not be possible or necessary

---

> ### Author Response · Authors · 2024-11-24
> **Response by Authors (1/8)**
>
> We appreciate the positive feedback highlighting that some of our observations are insightful and have the potential to contribute to a deeper understanding of LLVMs.
>
> ---
>
> **Q1.** The article focus on the LLaVA family. Could the authors explain their rationale for focusing on the LLaVA family and discuss potential limitations in generalizing these results to other LVLM architectures?
>
> **A1.** Thank you for your valuable feedback regarding the generalizability of our findings. We acknowledge the reviewer’s concern, especially given the recent proliferation of LVLMs with diverse architectures and functionalities. Models such as Qwen2-VL, LLaMA-Vision, TroL, Phantom, MoAI, Pangea, and Fuyu each introduce unique characteristics. For example:
>
> - **Qwen2-VL** demonstrates exceptional performance, supports multi-image inputs, arbitrary image resolutions, videos, and multilingual capabilities.
> - **Pangea** is an open-source multilingual LVLM trained on a 39-language visual instruction tuning dataset.
> - **LLaMA-Vision** is built on LLaMA-3.1 and incorporates advanced features.
> - **TroL** leverages layer traversal for training, while **Phantom** utilizes a DPO approach with dimension expansion.
> - **MoAI** integrates external computer vision modules (e.g., segmentation models).
> - **Fuyu** adopts a decoder-only transformer architecture without a specialized vision encoder, instead projecting image patches linearly into the first transformer layer.
>
> Despite these diverse advancements, most of these LVLMs have evolved based on the *model-stitching architecture* introduced by LLaVA. Given this, we focused on the LLaVA family for our experiments for the following reasons:
>
> 1. **Historical Significance**: The LLaVA model served as the foundation for recent LLVM developments, and many subsequent models have drawn inspiration from its architecture and approach.
> 2. **Relevance**: While older LLaVA models (e.g., LLaVA-1.5 and LLaVA-NeXT) are considered less competitive in terms of lower performance, they remain widely used as backbones for task-specific fine-tuning (e.g., Prometheus-Vision [1]) and serve as critical baselines in evaluation studies.
> 3. **Inclusion of Latest Models**: We also included LLaVA-OneVision in our study, as it achieves state-of-the-art performance among LLaVA-family models, ensuring that our experiments are not limited to outdated versions.
>
> However, we recognize the importance of validating our findings on other LVLM architectures. To address this concern, we conducted additional experiments on **Qwen2-VL**, a leading LVLM known for its robust performance and advanced features. Specifically, we investigated key aspects such as **permutation invariance**, **sensitivity to spatial structure**, and **occlusion** to demonstrate the generalizability of our observations beyond the LLaVA family.
>
> **Additional Results: Permutation Invariance (Table 4 in Appendix E.1.)**
>
> | LLVMs | MMVP | Q-Bench | MME | MMStar | MM-Vet | LLaVA$^W$ | MathVista | SQA$^I$ | ChartQA | AI2D | Avg. Δ |
> | --- | --- | --- | --- | --- | --- | --- | --- | --- | --- | --- | --- |
> | LLaVA-1.5 | 34.67 | 59.73 | 1850.07 | 34.20 | 31.50 | 67.50 | 24.70 | 65.59 | 16.92 | 53.34 | - |
> | + Perm. | 36.00 (🔼 1.33) | 59.60 (🔻 0.13) | 1874.60 (🔼 24.53) | 33.33 (🔻 0.87) | 30.40 (🔻 1.10) | 66.20 (🔻 1.30) | 21.20 (🔻 3.50) | 65.44 (🔻 0.15) | 14.08 (🔻 2.84) | 52.69 (🔻 0.65) | 🔻 0.59 |
> | LLaVA-NeXT | 36.67 | 63.55 | 1874.42 | 37.80 | 43.50 | 75.50 | 32.00 | 62.12 | 66.06 | 64.02 | - |
> | + Perm. | 37.33 (🔼 0.67) | 62.54 (🔻 1.00) | 1890.19 (🔼 15.78) | 36.87 (🔻 0.93) | 43.40 (🔻 0.10) | 75.80 (🔻 0.30) | 21.70 (🔻 10.30) | 62.12 (🔻 0.00) | 34.55 (🔻 31.51) | 64.02 (🔻 0.00) | 🔻 2.71 |
> | LLaVA-OneVision | 60.67 | 77.26 | 1982.50 | 59.87 | 57.80 | 87.40 | 61.80 | 94.00 | 93.52 | 81.25 | - |
> | + Perm. | 59.33 (🔻 1.33) | 76.99 (🔻 0.27) | 1964.30 (🔻 18.20) | 54.93 (🔻 4.93) | 47.60 (🔻 10.20) | 82.30 (🔻 5.10) | 53.50 (🔻 8.30) | 89.24 (🔻 4.76) | 58.26 (🔻 35.26) | 75.58 (🔻 5.67) | 🔻 9.40 |
> | Qwen2-VL-7B | 50.67 | 77.06 | 2356.70 | 55.27 | 62.60 | 94.10 | 59.80 | 0.00 | 94.83 | 80.21 | - |
> | + Perm. | 48.67(🔻 2.00) | 77.19 (🔼 0.13) | 2266.96 (🔻 89.74) | 53.47 (🔻 1.80) | 62.20 (🔻 0.40) | 93.20 (🔻 0.90) | 53.10 (🔻 6.70) | 0.00 (🔻 0.00) | 83.59 (🔻 11.25) | 77.43 (🔻 2.78) | 🔻 12.82 |
> - *Note: Due to the large sample size of the SQA-IMG dataset, we have not yet completed the experiment for Qwen2-VL, but will include it in our final paper.*

---

> ### Author Response · Authors · 2024-11-24
> **Response by Authors (2/8)**
>
> We investigate the extent to which other LVLMs exhibit permutation invariance under the same experimental settings described in Table 1. Overall, the Qwen2-VL-7B model demonstrate permutation invariance on average, displaying patterns similar to those observed in the LLaVA-family models. A more detailed analysis across benchmarks reveals interesting patterns. In perception-focused benchmarks, such as MMVP, Q-Bench, MME, and MMStar (the latter two being integrated capability benchmarks that include perception-related tasks), the performance drop due to permutation is negligible. However, in text-rich benchmarks like MathVista and ChartQA, the performance drops significantly. **These benchmarks require an understanding of detailed numerical information and highly structured geometric graphs, where maintaining the spatial structure of visual patch tokens is critical.**
>
> **Additional Results: Sensitivity to Spatial Structures (Figure 11 in Appendix E.2.)**
>
> We randomly shuffled image patches to evaluate their impact on model performance and observed that Qwen2-VL-7B exhibits a similar tendency to LLaVA-family models. Specifically, we found that Qwen2-VL and LLaVA-OneVision are highly sensitive to spatial structures in text-rich benchmarks (e.g., MathVista, AI2D), which contain detailed numerical information. Notably, the performance of the Qwen2-VL model dropped significantly when the grid size is 2. To understand why Qwen2-VL is particularly sensitive, we hypothesize that this behavior is linked to its use of enhanced multi-modal rotary position embeddings (M-ROPE) [3]. This embedding mechanism likely contributes to the performance degradation observed when image patches are shuffled. Conversely, the model is relatively insensitive to spatial structures in perception-centric benchmarks (e.g., MMVP). We speculate that this insensitivity arises from the robustness of vision encoders, as reported in a prior study [4], which suggest that ViT-based models are  robust to spatial disruptions in datasets like ImageNet. However, text-rich images, unlike perception-related images, include intricate numerical details. **Shuffling image patches in such cases disrupts geometric relationships or relative magnitudes in plots or charts, making accurate interpretation of these images significantly more challenging.**
>
> **Additional Results: Occlusions (Figure 12 in Appendix E.3.)**
>
> In Figure 12, we observe that the Qwen2-VL-7B model exhibits a similar tendency to the LLaVA family models. Notably, the performance trend slope of the Qwen2-VL-7B model closely resembles that of LLaVA-OneVision, suggesting that both models — currently high-performing LVLMs — share similar patterns. This alignment supports the generalizability of our observations. Specifically, LVLMs demonstrate relatively strong performance under occlusions. For instance, in the AI2D dataset, even when 50–70% of image patches are missing, the models can still provide correct answers to some extent. Moreover, in these scenarios, the Qwen2-VL and LLaVA-OneVision models outperform LLaVA-1.5 and LLaVA-NeXT, even when no patches are missing. These results indicate that state-of-the-art LVLMs possess strong visual understanding capabilities. This suggests that improving visual understanding during training contributes significantly to high performance and robustness against occlusion.
>
> Here, we aim to demonstrate “model-side generalizability” in terms of permutation invariance, sensitivity to spatial structure, and occlusion. Overall, the Qwen2-VL model exhibits similar patterns across all three experiments, suggesting that our findings generalize to other LVLMs. However, as mentioned earlier, there is a wide variety of LVLMs in this field. To address this, we plan to continuously update our work by conducting additional experiments on other LVLMs to further evaluate the generalizability of our findings. We appreciate the reviewer’s comment and are committed to addressing it.

---

> ### Author Response · Authors · 2024-11-24
> **Response by Authors (3/8)**
>
> **Q2.** In Section 3.3, how is the grouping done to quantify local information in patches? How are neighbours defined, and what is the impact of this definition on the results?
>
> **A2.** In Section 3.3, we group adjacent patches together. For clarity, we illustrate this grouping method in Figure 13 in Appendix E.4. Similar to a convolution layer's operation, we combine neighboring patches into single groups (marked by matching colors) before inputting them into the model. By adjusting the grid size, which determines how many patches form each group, we examine whether this affects the pattern shown in Figure 1. We conducted additional experiments using a 3 × 3 grid of patches and present the visualization in Appendix E.4. We found that increasing the number of grid patches leads to more precise observations. Compared to a 6 × 6 grid of patches, a 3 × 3 grid yields less precise observations. While conducting experiments on all visual patch tokens (576 for LLaVA-1.5) would provide the most precise interpretations, this approach is computationally intensive, as mentioned in Section 3.3. Therefore, we believe our chosen grid size strikes a reasonable balance for obtaining meaningful interpretations.
>
> ---
>
> **Q3.** The explanation of backpropagation and loss signal coming from the assistant in line 234 is unclear. Can the authors provide more details or clarify the average performance drop, which seems to be benchmark-dependent?
>
> **A3.** Thank you for the thoughtful feedback about clarifying our claims and observations. Our analysis reveals that "permutation invariance" is both benchmark-dependent and capability-dependent. For perception-related benchmarks like MMVP and Q-Bench, the performance drop is minimal—LLaVA-1.5 and LLaVA-NeXT even show slight improvements in some cases. However, text-rich benchmarks requiring reasoning capabilities (such as MathVista and ChartQA) show significant performance drops. **These benchmarks require understanding detailed numerical information and complex geometric graphs, which makes the spatial structure of visual patch tokens crucial.** Based on these observations, we slightly modify the original findings by adding "depending on the benchmark" in Section 3.3 (highlighted in blue).
>
> During visual instruction tuning, LLVMs learn from datasets of paired images and user-assistant conversations. The training process involves calculating cross-entropy loss on the assistant's textual outputs and backpropagating this loss signal. However, there's a significant imbalance: while models process many visual tokens (576 for LLaVA-1.5 and over 1,000 for Qwen2-VL), the assistant's output contains relatively few text tokens. This imbalance means that **using only the loss signal from assistant text tokens may not adequately optimize all visual patch tokens, especially those at the edges.** Since each visual patch token contains localized information (as shown in Figure 1), this **training approach likely causes the model to focus on more informative, critical visual patches** while giving less attention to peripheral ones. We therefore hypothesize that this "permutation invariance" emerges because recent LLVMs are trained through backpropagation, where the loss signal primarily comes from the assistant's textual outputs.

---

> > ### Comment · Reviewer_v3XH · 2024-11-26
> > **permutation invariance explanation**
> >
> > Thank you for your answers, I apologize for the delay, I'm going through them. I'm still having trouble to understand your point/interpretation and especially "only using the loss signal from the assistant text tokens may not adequately optimize all visual patch tokens" :
> > How the imbalance between the number of visual tokens and output text tokens inform you about the visual tokens gradient coverage? All the patches get gradients, it's just that some of them may not be informative.
> > I think that because it's specifically depend on benchmarks it might just mean that
> > 1. some benchmark do not need information about specific patches order and rather test the presence of global information (that might already be encoded in a lot of patches)
> > 2. most of edge patches are not as informative as the other ones (as can be seen in Figure 1)

---

> ### Author Response · Authors · 2024-11-24
> **Response by Authors (4/8)**
>
> **Q4.** In Table 2, what setting is used to measure the frequency of outputting 1 (synthetic or original)? Is there a difference between these cases? Can the authors also report the frequency of outputting "no" and precision/recall for yes/no questions?
>
> **A4.** We measure the frequency of outputting 1 is in the “synthetic” cases. We additionally report the frequency of both synthetic and original cases as follows.
>
> | Model | Synthetic | Original |
> | --- | --- | --- |
> | LLaVA-1.5 | 81.0 | 44.4 |
> | LLaVA-NeXT | 50.0 | 13.8 |
> | Meteor | 9.5 | 7.5 |
> | LLaVA-OneVision | 12.0 | 8.3 |
>
> As shown in the above Table, in overall, the “original” ratio of LLVM generating “1” in free-form question types is reduced compared to the “synthetic” cases. This results suggest that LLVMs can effectively interpret and understand the detailed numerical information in the given image, thereby, the phenomenon that LLVM tend to use their commonsense reasoning is reduced. However, considering the ratio of LLaVA-1.5 (44%), this ratio is not negligible. Therefore, in the future, we need to build more challenging benchmark that do not rely on the commonsense reasoning.
>
> In addition, we report the frequency of outputting “no” in the “synthetic” case and the precision/recall for yes/no questions as follows:
>
> | Model | No (%) |
> | --- | --- |
> | LLaVA-1.5 | 64.0 |
> | LLaVA-NeXT | 54.4 |
> | Meteor | 82.8 |
> | LLaVA-OneVision | 70.5 |
>
> | Model | Precision (Synthetic) | Recall (Synthetic) | Precision (Original) | Recall (Original) |
> | --- | --- | --- | --- | --- |
> | LLaVA-1.5 | 49.2 | 36.8 | 59.4 | 43.7 |
> | LLaVA-NeXT | 50.0 | 47.1 | 54.0 | 39.1 |
> | Meteor | 55.2 | 18.4 | 78.0 | 36.8 |
> | LLaVA-OneVision | 54.9 | 32.2 | 72.4 | 72.4 |
>
> We observe that most LVLMs prefer to answer “no” for yes/no question types in multiple-choice question (MCQ) formats. This suggests that, when presented with synthesized images, LVLMs struggle to solve the given questions effectively. **Instead of attempting to provide an answer based on the limited or unclear information available in the synthesized images, LVLMs tend to decline by answering “no,” leading to an increased frequency of “no” responses compared to “yes.”**
>
> Furthermore, across all models, the **original dataset consistently yields better performance in both precision and recall**. This indicates that LVLMs face significant challenges in solving questions based on synthesized information. In the “synthetic” case, **precision is consistently higher than recall**, reflecting the tendency of LVLMs to output “no” answers more frequently than “yes” answers. This behavior underscores the challenges LVLMs face in effectively using synthesized visual information to provide accurate answers to yes/no questions.
>
> ---
>
> **Q5.** Evaluating the loss of cross-modal alignment is interesting. However the evaluation method used in Table 3 is unclear and may not be fair compared to a contrastive approach. Could the authors give more details about their evaluation setup (prompt used?) as it is known that prompt optimization impacts even zero-shot CLIP models classification performance. Can the authors consider alternative approaches, such as log-likelihood comparison of description prompts (e.g., "Describe the image: This is an image of a [class label]")? As it is done in captioning models evaluation for instance?
>
> **A5.** To evaluate CLIP models on zero-shot classification tasks, we use the prompt templates provided by [CLIP-Benchmark](https://huggingface.co/clip-benchmark). All prompt templates are presented in Appendix E.6. For LLVMs, we use the same prompt templates but include the label candidates. For example, with the Food101 dataset, we provide the prompt: "*What is the type of food in the image? Please answer only a single type of food in {class_labels}.*" Here, {class_labels} represents the provided label candidates from each dataset. To evaluate the LLVM's generated answer, we use ChatGPT with the prompt: "*Please only answer the question in yes or no. Is the "Prediction" correctly predicting the right "Label"? Label: {label}; Prediction: {outputs}.*" This evaluation method follows the approach used in study [2]. Following the reviewer's suggestion, we conducted an additional experiment using "log-likelihood" with the prompt "*Describe the image: This is an image of a [class label].*" However, this approach yielded less meaningful results, with accuracy below 5%, on most datasets. We believe this low performance stems from the large number of labels (over 100 on average) in the image datasets, making log-likelihood comparison challenging. Even when considering Top-5 accuracy, performance only reached about 10%. While this is a valid method for image classification tasks, our direct label prediction approach using provided candidates (following study [2]) proved more effective. We appreciate the reviewer's suggestion of this alternative approach.

---

> ### Author Response · Authors · 2024-11-24
> **Response by Authors (5/8)**
>
> **Q6.** How is the 1k subset obtained in Table 3?
>
> **A6.** We randomly sampled 1,000 examples from each original dataset. Dataset sizes vary—for example, the Food101 test set contains 25.3K samples. To reduce computational costs while ensuring reliable results, we conducted five experiments with different random seeds and report both the average accuracy and standard deviation.
>
> We report the **Top-1 accuracy (%)**with standard deviation (in parentheses) for LLVMs and their corresponding vision encoder models on 1K-sample subsets from Caltech100, CIFAR-100, Food101, Pets, Country211, EuroSAT, and AirCraft. Results are averaged across five experiments.
>
> | LLVMs | Caltech101 | CIFAR-100 | Food101 | Pets | Country211 | EuroSAT | AirCraft | Avg. |
> | --- | --- | --- | --- | --- | --- | --- | --- | --- |
> | **CLIP ViT 336** | 84.50 (0.52) | 75.10 (1.74) | 93.72 (0.61) | 93.48 (0.63) | 31.14 (0.84) | 58.90 (0.84) | 34.08 (0.77) | 67.27 (0.85) |
> | **LLaVA-1.5** | 43.76 (2.69) (🔻40.74) | 48.36 (2.47) (🔻26.74) | 57.22 (0.61) (🔻36.5) | 73.22 (0.37) (🔻20.26) | 12.20 (0.47) (🔻18.94) | 11.72 (0.34) (🔻47.18) | 17.00 (0.72) (🔻17.08) | 37.64 (1.10) (🔻29.63) |
> | **CLIP ViT 14** | 84.52 (0.56) | 75.86 (1.06) | 92.78 (0.41) | 93.08 (0.30) | 28.68 (1.44) | 58.46 (0.80) | 32.98 (1.02) | 66.62 (0.80) |
> | **LLaVA-NeXT** | 56.68 (2.29) (🔻27.84) | 45.36 (1.02) (🔻30.5) | 53.14 (1.00) (🔻39.64) | 75.06 (0.93) (🔻18.02) | 12.94 (0.35) (🔻15.74) | 8.34 (0.59) (🔻50.12) | 12.66 (0.46) (🔻20.32) | 37.74 (🔻28.88) |
>
> As shown in the table above, even with different random seeds, the results consistently demonstrate that LLVMs struggle to preserve their original visual understanding capability. The small standard deviations indicate that these results are reliable. We update Table 3 with the above results.
>
> ---
>
> **Q7.** The section on shared world concepts is hard to follow. What metric is used to measure alignment, and why was the DOCCI dataset chosen? How is alignment measured with long captions given CLIP's 77-tokens context?
>
> **A7.** Thank you for pointing out that the section on shared world concepts is difficult to follow due to the lack of explanation about how alignment is measured. To address this, we have provided detailed clarification below, which has also been added to Appendix B.
>
> ### **Alignment Metric**
>
> To evaluate the alignment between representations from two models, we employ the **Mutual k-Nearest Neighbor (MNN) Metric**. This metric assesses local similarity by computing the intersection of the k-nearest neighbor sets for each sample from the two models' representation spaces. The alignment is then quantified based on the size of these intersections. While the detailed mathematical formulation is provided in the Appendix B, we describe the procedure here in simpler terms:
>
> 1. **Feature Extraction:** Extract features from the models for each sample in the dataset.
> 2. **k-Nearest Neighbor Sets:** Compute the k-nearest neighbor sets for each extracted feature.
> 3. **Pairwise Alignment:** Measure alignment for a pair of features as the normalized size of the intersection of their k-nearest neighbor sets.
> 4. **Average Alignment:** Compute the average alignment across all samples in the dataset.

---

> ### Author Response · Authors · 2024-11-24
> **Response by Authors (6/8)**
>
> ### **How to use in our experiment**
>
> To assess the alignment between a suite of large language models (LLMs) and vision models, we utilize the image-caption pair dataset DOCCI. Specifically, in DOCCI, the dataset consists of image-caption pairs $D = \{(x_i, y_i)\}_{i=1}^{|D|},$ where $x_i$ denotes the image and $y_i$ denotes the corresponding caption text.
>
> For our experiment, we prepare three models: an LLM ($f_L$), a vision encoder from a vision-language model without visual instruction tuning ($f_V$), and a vision encoder with a projector, representing a vision-language model with visual instruction tuning ($f_{VP}$). The vision encoder in $f_{VP}$ is kept identical to $f_V$. For example, `CLIP-L/336px` is used as the vision encoder for both $f_V$ and $f_{VP}$ when paired with `LLaVA-1.5`.
>
> In our experiment, we explore the degree of alignment lost after visual instruction tuning, guided by the Platonic representation hypothesis. We assume that in a successful LLVM, the projector should effectively represent the visual world and enable the LLM to understand and interpret the given image accurately. We calculate two alignment scores: one between $f_L$ and $f_V$, and another between $f_L$ and $f_{VP}$. The discrepancy between these scores reflects the extent to which alignment performance deteriorates.
>
> To compute the alignment scores, we follow these steps:
>
> 1. Extract features from $f_L$ by providing the input text $y_i$. We then apply average pooling to all the extracted hidden states.
> 2. Extract features from $f_V$ by providing the image $x_i$, using only the feature corresponding to the `[CLS]` token.
> 3. Extract features from $f_{VP}$ by providing the image $x_i$, applying average pooling to all visual patch tokens (e.g., 576 tokens in `LLaVA-1.5`) produced by the projector.
>
> Finally, we calculate the alignment scores using these extracted features via the mutual nearest-neighbor alignment metric.
>
> Regarding the reviewer's concern about "long captions given CLIP's 77-tokens context," this limitation is not an issue in our alignment calculations. In our experiments, we **do not use CLIP's text encoder to process captions; instead, we rely on the LLM ($f_L$) to extract text features.** This misunderstanding may have arisen due to our omission of details about the alignment measurement process, and we sincerely apologize for the confusion.
>
> ### **Motivation behind Selecting the DOCCI Dataset**
>
> While complex image understanding (e.g., charts, mathematical representations, code snippets, and diagrams) is crucial for creating a helpful assistant, we believe that an LLVM should first master natural scene comprehension to become a broadly applicable personal AI assistant—particularly for applications like smart glasses (e.g. Meta AI's glasses [5]) or real-time cameras (e.g., Project Astra). Effective alignment between language and vision modalities requires paired datasets with detailed image captions that capture essential visual features: attributes, spatial relationships, object counts, objects, text rendering, viewpoints, optical effects, and world knowledge. Following these requirements, we sought a dataset that emphasizes (1) natural scenes and (2) highly descriptive captions. The DOCCI dataset effectively meets these criteria. Other potential datasets include Localized Narratives, CC12M, COCO-Caption, WIT, or RedCaps12M. We plan to conduct additional experiments with these datasets in future work to enhance the generalizability of our observations.
>
> ---
>
> **Q8.** Are the local scores in Figure 8 averaged over the whole dataset or just one example? Same question for Figure 1 ?
>
> **A8.** The results reported in both Figure 1 and Figure 8 are averaged across the entire dataset. In Figure 1, we calculate the patch information loss (PIL) for each group-wise visual patch token and average these values across all samples. In Figure 8, we introduce noise—randomly sampled from a uniform distribution as described in Equation (2)—into each group-wise visual patch token embedding, targeting specific visual patch token indices (out of 36 indices). The importance scores are then averaged across all samples in the dataset. This averaging ensures our results reflect general patterns rather than individual examples.

---

> > ### Comment · Reviewer_v3XH · 2024-11-26
> > **alignment metric**
> >
> > Thank you for your answer and the definition of how you compute the alignment in Table 7. I think i understood now how you compute the alignment metric but feel free to correct me. One thing is still bugging me though : why don't you also compute the alignment for CLIP before instruction finetuning  (so your $f_V$ right?) with the average pooling of the patches and rather choose the CLS token  to make it comparable with how you compute the visual representation of LLaVa using $f_{VP}$? it might not be fair since the CLS might be more aligned that the rest of the patches with the text representations.

---

> ### Author Response · Authors · 2024-11-24
> **Response by Authors (7/8)**
>
> **Q9.** Can the authors comment on the choices of datasets for different parts of the analysis (e.g., MMVet for localization of information, MMStar for layers importance)? Are these results model-, dataset-, or sample-specific, and how do they impact the generalizability of the findings? I know this question is quite general but the justification is not clear to me and the drawn conclusions seem to be a general so I would except such visualizations to be performed on a subset of the different datasets.
>
> **A9.** Let us explain our dataset choices for different parts of the analysis. For importance score calculation, we aimed to use benchmarks that cover "integrated capabilities," such as MME, MMStar, and MM-Vet. Computing the importance score is highly time-intensive. For each run, we calculate importance scores for each group-wise position (36 positions for LLaVA-1.5) and repeat the experiment K times (K=10), resulting in 360 experiments per benchmark. Similarly, calculating layer importance is computationally demanding. Therefore, based on dataset size, we selected MM-Vet (219 samples) and MMStar (1.5K samples) for initial experiments, as MME contains 2.37K samples. We acknowledge that computational intensity is a limitation of our "importance score" metric, and to improve generalizability, we need to conduct more experiments across other LLVMs and datasets. We will continue to update these results as they become available.
>
> We conducted additional experiments on the MME dataset, shown in Figure 14 in Appendix E.8. (We focused on evaluating LLVMs on the MME dataset since MM-Vet requires OpenAI API calls, which are currently beyond our budget. We plan to add more results on different datasets in the future.) Figure 14 (left) shows that the lower layers (< 10) play a crucial role in handling integrated capabilities. Figure 14 (right) demonstrates that in the lower layers (< 12), the image modality is more important than the text modality. **The tendencies observed in the MME dataset align with those in the MMStar dataset,** as shown in Figure 9. However, a key difference is the layer index at which the modality importance shifts—this transition occurs at a higher layer index in the MME dataset. We hypothesize that LLVMs dedicate more effort to image comprehension in the MME dataset compared to the MMStar dataset. While this might be because MME dataset images are more challenging to comprehend, we cannot confirm this hypothesis without further investigation in future studies.
>
> ---
>
> **Q10.** The conclusion drawn from Figure 9 lacks a clear explanation of how the importance score is obtained. Can the authors provide more details on the method used (e.g., noised input or weights, which proportion)?
>
> **A10.** Thank you for highlighting the lack of clarity in the explanation of how the importance score is obtained. We have now provided a detailed explanation, which has been added to Appendix C, as outlined below:
>
> Based on Equation (2), we prepare the constraint candidate set $\mathcal{C}_t$, defined as a squared boundary: $-\epsilon + |x_t| \leq z_t \leq \epsilon + |x_t|$, where $\epsilon \sim \text{Uniform}(-1, 1)$. At each iteration, we randomly sample a noise vector $z_t$ and apply it to the target component. Below, we detail how this is done for visual tokens, layers, and modalities.
>
> - **Visual Token Importance:** When evaluating the importance of a visual patch token, the noise vector is injected into the group-wise visual patch token embeddings at the target position. For instance, we examine 36 positions. To measure the importance of position 0, we add the noise vector to the corresponding visual patch token embeddings at position 0, while leaving all other patch token embeddings unchanged. These modified embeddings are then input into the LLM for further processing.
> - **Layer-Wise Importance:** To explore layer-wise importance, the noise vector is injected into the target layer before it is processed by the LLM. Specifically, the noise is applied directly to the layer's input embeddings before passing the target layer, ensuring that the perturbation affects only the selected layer.
> - **Modality Importance:** To calculate the importance of the textual modality (IT), the noise vector is injected only into the positions corresponding to text inputs within the target layer, while leaving the positions associated with visual patch tokens unchanged. Conversely, for visual modality importance (II), the noise vector is injected into the positions corresponding to visual patch tokens within the target layer. The relative importance score for each modality is then computed as II/IT.
>
> To enable better interpretation across layers, all importance scores (both layer-wise and modality-specific) are normalized using min-max normalization.

---

> ### Author Response · Authors · 2024-11-24
> **Response by Authors (8/8)**
>
> Based on the reviewer's thoughtful suggestions to make our observations more rigorous and reliable, we have incorporated the feedback into our revised paper. These additions and modifications are highlighted in blue for clarity: (1) We provide limitations and future directions in Appendix A and show additional results in Appendix E for generalizability, (2) We explain our rationale for selecting the DOCCI dataset, and (3) To further increase the generalizability of our findings, we acknowledge the need for additional experiments on other datasets (however, due to budget and resource constraints, we were unable to conduct extensive experiments, though we will continue to update and add results). We greatly appreciate the reviewer's meaningful and helpful suggestions for improving our paper.
>
> ---
>
> **References**
>
> [1] Lee, Seongyun, et al. "Prometheusvision: Vision-language model as a judge for fine-grained evaluation." *arXiv preprint arXiv:2401.06591* (2024).
>
> [2] Zhai, Yuexiang, et al. "Investigating the catastrophic forgetting in multimodal large language models." *arXiv preprint arXiv:2309.10313* (2023).
>
> [3] Wang, Peng, et al. "Qwen2-vl: Enhancing vision-language model's perception of the world at any resolution." *arXiv preprint arXiv:2409.12191* (2024).
>
> [4] Naseer, Muhammad Muzammal, et al. "Intriguing properties of vision transformers." *Advances in Neural Information Processing Systems* 34 (2021): 23296-23308.
>
> [5] https://www.meta.com/smart-glasses/
>
> [6] https://www.youtube.com/watch?v=nXVvvRhiGjI

---

> ### Author Response · Authors · 2024-11-28
> **Response by Authors (permutation invariance explanation)**
>
> Thank you for thoroughly reviewing our responses and for raising these thoughtful points. We sincerely appreciate your efforts in engaging with our work.
>
> To address your question, we acknowledge and agree with your insightful interpretation regarding **"permutation invariance being benchmark-dependent."** Regarding the statement **"only using the loss signal from the assistant text tokens may not adequately optimize all visual patch tokens,"** we recognize that this was a speculative observation and not strongly substantiated in our work. Beyond the two reasons you mentioned (1 and 2), we aimed to explore potential factors in the training process that might contribute to the observed permutation invariance phenomenon.
>
> One preliminary hypothesis we considered is whether the imbalance between the number of visual tokens and text tokens affects the model’s ability to optimize visual localization. For example, in LLaVA-1.5, there are 576 visual tokens, while the number of text tokens is roughly five times smaller (e.g., in LLaVA-Instruct-150K). While it is true that **"all patches receive gradients,"** we hypothesize that some visual patches—particularly those corresponding to less salient regions—may receive weaker gradient signals during training due to their relatively lower contribution to the output text (assistant-side response). This could lead to under-optimization of certain patches, potentially affecting tasks that rely on fine-grained visual understanding. However, we acknowledge that this is a preliminary hypothesis and that proving it would require a more robust theoretical framework and empirical evidence.
>
> We strongly agree with your interpretation that **"permutation invariance is benchmark-dependent"** and that **"most edge patches are less informative than others,"** as illustrated in Figure 1. However, we also believe that our hypothesis presents a plausible contributing factor. Once again, we sincerely thank you for your thoughtful feedback. Your insights align with and enrich our understanding, encouraging us to refine and clarify our arguments further.

---

> ### Author Response · Authors · 2024-11-28
> **Response by Authors (alignment metric)**
>
> Thank you for raising this important point about ensuring a **"fair comparison."** We appreciate your thoughtful observation. In our original approach, we followed the default setting from prior work [1] to compute alignment and did not compare $f_V$ using average pooling of the patches. However, as you pointed out, relying solely on the CLS token for $f_V$ might not be entirely fair. To address this concern, we conducted an additional experiment to ensure a fair comparison. Specifically, when computing alignment, we **extracted features using average pooling of the patches instead of the CLS token.** The results of this experiment are presented in **the below Table.** From this analysis, we **observed the same tendency:** alignment is not well preserved after instruction tuning. This finding reinforces our hypothesis that the degradation in alignment preservation could be linked to the role of the projector. We sincerely thank you for this insightful suggestion, which has helped us strengthen the fairness and robustness of our comparison.
>
> | Model          | bloom 0.55b | bloom 1.1b | bloom 1.7b | bloom 3b | bloom 7b | olmo 1b | olmo 7b | gemma 2b | gemma 7b | vicuna 1.5 7b |
> |----------------|-------------|------------|------------|----------|----------|---------|---------|----------|----------|---------------|
> | CLIP-L/336px            | 0.2114      | 0.2150     | 0.2262     | 0.2340   | 0.2396   | 0.2322  | 0.2416  | 0.2374   | 0.2490   | 0.2644        |
> | LLaVA-1.5              | 0.1620      | 0.1640     | 0.1688     | 0.1706   | 0.1762   | 0.1686  | 0.1726  | 0.1734   | 0.1816   | 0.1874        |
> | CLIP-L/14              | 0.2250      | 0.2324     | 0.2436     | 0.2534   | 0.2574   | 0.2504  | 0.2622  | 0.2600   | 0.2770   | 0.2922        |
> | LLaVA-NeXT              | 0.1812      | 0.1836     | 0.1902     | 0.2002   | 0.1990   | 0.1986  | 0.2068  | 0.1958   | 0.2128   | 0.2262        |
>
> ---
>
> **Reference**
>
> [1] Huh, Minyoung, et al. "The platonic representation hypothesis." *arXiv preprint arXiv:2405.07987* (2024).

---

### Official Review · Reviewer_LdVk · 2024-11-03

**Soundness:** 2
**Presentation:** 2
**Contribution:** 2
**Rating:** 5
**Confidence:** 3

**Summary:**

This paper looks at how Large Language and Vision Models (LLVMs) like LLaVA handle different tasks. It finds that LLVMs don’t rely much on image order, sometimes solve math without exact details, and lose some visual skills when fine-tuned for language tasks. The study also notes that the first layers process visuals best. The authors suggest building tougher benchmarks to better test and improve these models.

**Strengths:**

This paper is useful because it digs into how large language and vision models (LLVMs) really work with visual information. It shows that LLVMs are flexible, able to handle scrambled image pieces and solve math problems without all the visual details. It also highlights that when LLVMs are tuned for complex reasoning, they lose some basic visual skills. These findings can help make LLVMs better by balancing complex reasoning with simpler perception tasks, potentially guiding the creation of new, stronger models.

**Weaknesses:**

The paper would benefit from more precise and carefully scoped conclusions. While the experiments provide interesting observations, the claims drawn from them are often overly broad. For example: 1) The claim about permutation invariance is based on VQA tasks, but this alone cannot support a general conclusion about LLVMs' visual processing capabilities.
2) The benchmarks used don't adequately test basic visual skills to support such sweeping statements about visual understanding

Specificity Needed:
Each conclusion needs clear boundaries about when it applies and when it doesn't.
The paper makes approximately 10 major claims, but lacks sufficient context about their limitations and specific application scenarios.
More precise scoping of these findings would make them more actionable for future research.
Currently, the broad generalizations make it difficult for other researchers to build upon these results effectively.

Minor points:
Figure 5's text is too small to read effectively, hampering understanding of key results.

**Questions:**

Besides the weakness, to improve, I suggest the authors:

1. Clearly specify the conditions under which each conclusion holds
2. Acknowledge the limitations of their experimental setup
3. Provide more nuanced interpretations of their results
4. Consider additional experiments to support broader claims

---

> ### Author Response · Authors · 2024-11-24
> **Response by Authors (1/2)**
>
> We appreciate the positive feedback regarding our paper's usefulness and several interesting findings.
>
> ---
>
> **Q1.** The paper would benefit from more precise and carefully scoped conclusions. While the experiments provide interesting observations, the claims drawn from them are often overly broad. For example: 1) The claim about permutation invariance is based on VQA tasks, but this alone cannot support a general conclusion about LLVMs' visual processing capabilities. 2) The benchmarks used don't adequately test basic visual skills to support such sweeping statements about visual understanding
>
> **A1.** Thank you for the thoughtful feedback regarding the **broadness of our claims**. As the reviewer pointed out, the evaluation benchmarks used in our experiments primarily consist of VQA tasks, which focus on binary, multiple-choice, and free-form question types. To address whether our claim regarding **"permutation invariance" generalizes to other datasets**, we have considered additional tasks such as image captioning, image retrieval, and referring expression comprehension. In this rebuttal, we conduct additional experiments using **image captioning tasks to evaluate the generalizability of our claims.** In the future, we plan to include experiments on other tasks. These tasks inherently require "visual processing capabilities," such as understanding attributes, viewpoints, scenes, and objects.
>
> For this investigation, we evaluated three standard datasets: COCO-Captions (Karpathy test set), NoCaps (validation set), and TextCaps (validation set). To generate captions, we followed the default prompting setup from `LMMs-Eval`, which uses the prompt: "*Please carefully observe the image and come up with a caption for the image.*" We employed standard evaluation metrics—ROUGE-L and CIDEr—to assess performance. The results are summarized below:
>
> | **LLMs** | **COCO-Captions ROUGE-L** | **COCO-Captions CIDEr** | **NoCaps ROUGE-L** | **NoCaps CIDEr** | **TextCaps ROUGE-L** | **TextCaps CIDEr** |
> | --- | --- | --- | --- | --- | --- | --- |
> | **LLaVA-1.5** | 22.01 | 0.97 | 25.34 | 1.52 | 22.46 | 6.09 |
> | + **Perm.** | 22.62 (🔼 *0.62)* | 1.26 (🔼 *0.29)* | 26.05 (🔼 *0.71)* | 2.89 (🔼 *1.37)* | 22.94 (🔻 *0.48)* | 7.28 (🔼 *1.19)* |
> | **LLaVA-NeXT** | 21.63 | 8.12 | 22.78 | 6.26 | 21.49 | 15.94 |
> | + **Perm.** | 21.86 (🔼 *0.24)* | 7.64 (🔻 *0.48)* | 22.68 (🔻 *0.10)* | 5.81 (🔻 *0.44)* | 20.19 (🔻 *1.29)* | 12.30 (🔻 *3.65)* |
> | **LLaVA-OneVision** | 57.23 | 116.25 | 56.09 | 86.60 | 44.58 | 72.69 |
> | + **Perm.** | 56.70 (🔻 *0.53)* | 116.17 (🔻 *0.08)* | 56.36 (🔼 *0.26)* | 85.94 (🔻 *0.66)* | 44.19 (🔻 *0.39)* | 68.18 (🔻 *4.52)* |
> | **Qwen2-VL** | 39.98 | 44.61 | 44.01 | 39.37 | 35.80 | 46.86 |
> | + **Perm.** | 37.19 (🔻 *2.79)* | 39.29 (🔻 *5.33)* | 42.70 (🔻 *1.31)* | 38.35 (🔻 *1.02)* | 35.31 (🔻 *0.49)* | 44.64 (🔻 *2.22)* |
>
> From the above results, we observe similar trends across image captioning datasets: **most LLMs exhibit "permutation invariance."** Interestingly, on the TextCaps dataset, the performance drop is more pronounced compared to other datasets, suggesting relatively greater "permutation variance." TextCaps contains more complex images (e.g., those with detailed numerical information) compared to the other datasets, which may explain this phenomenon.
>
> When comparing these findings to those in Table 1 of our paper, we note that in perception-related tasks (e.g., involving natural scenes), LLMs generally exhibit permutation invariance. However, in reasoning-related tasks (e.g., MathVista) involving images with complex structures (e.g., charts or diagrams), LLMs demonstrate greater permutation variance. This suggests that maintaining the geometric or positional structure of plots and charts is crucial.
>
> Based on these results, we refine our conclusion as follows: **LLMs exhibit "permutation invariance" in perception-related tasks**, such as image captioning involving natural scenes, but **demonstrate "permutation variance" in reasoning-related tasks that require processing complex structures.**
>
> We add this result into Appendix E.7.

---

> ### Author Response · Authors · 2024-11-24
> **Response by Authors (2/2)**
>
> **Q2.** Specificity Needed: Each conclusion needs clear boundaries about when it applies and when it doesn't. The paper makes approximately 10 major claims, but lacks sufficient context about their limitations and specific application scenarios. More precise scoping of these findings would make them more actionable for future research. Currently, the broad generalizations make it difficult for other researchers to build upon these results effectively.
>
> **A2.** Thank you for your thoughtful feedback regarding the need for increased specificity. For our detailed response to this point, please refer to the "Common Response" post.
>
> ---
>
> **Q3.** Figure 5's text is too small to read effectively, hampering understanding of key results.
>
> **A3.** In response to the comment, given the 10-page limit, we have included additional examples of synthesized and occluded images in Appendices G and I, respectively.
>
> ---
>
> Following the reviewer's suggestions, we have made several improvements to enhance the quality of our paper:
>
> 1. **Clear Specification of Experimental Conditions:** We now explicitly specify the conditions and assumptions under which our experiments were conducted. These details can be found in Appendix D.
> 2. **Acknowledgment of Limitations:** We provide a detailed discussion of the limitations of our experiments in Appendix A.1, ensuring transparency and clarity regarding the scope of our findings.
> 3. **Nuanced Interpretations:** We have added further interpretations of our results to provide deeper insights in the main paper (highlighted in blue for ease of identification) and in the Appendix E.
> 4. **Additional Experiments:** To support the generalizability of our findings, we have included additional experiments in Appendix E.
> 5. **Future Research Directions:** To offer meaningful insights for future research, we have also expanded the discussion section in Appendix A.2.
>
> We greatly appreciate the reviewer’s valuable suggestions, which have significantly helped us improve the paper.

---

### Author Response · Authors · 2024-11-24
**Common Response for Reviewers (1/9)**

We appreciate all reviewers' thoughtful feedback regarding the **broadness of our claims and the need to present limitations and future directions for each finding.** We acknowledge that many conclusions from our observations lacked the clarity needed to effectively guide future research. To address this, we have revisited each observation and provided more specific, well-defined conclusions. We have outlined the experimental limitations of each observation and highlighted practical application scenarios where these findings may be most beneficial. We have also proposed potential future research directions that build upon these observations. These clarifications and refinements are presented below (and shown in detail in Appendix A):

---

### **Findings 1: Each visual patch token encapsulates localized visual information.**

In this finding, we observe that each visual patch token from the projector captures a localized understanding of the patch area corresponding to its position in the image.

- **Limitations**: In our experiments, this observation was investigated solely on the LLaVA-1.5 model using two evaluation benchmarks: MMStar and MME. Both benchmarks primarily assess the integrated capabilities of LLVMs (e.g., perception and reasoning). This limited scope means that we cannot guarantee the generalizability of our findings, despite using widely recognized evaluation benchmarks and a popular LLVM model. Furthermore, more diverse and challenging benchmarks, as well as advanced and highly intelligent LLVMs, were not considered in our study. To reduce computational complexity, we simplified the evaluation by grouping neighboring visual patch tokens. While this approach streamlined the analysis, it does not verify the localized information within each visual patch token in detail. To address this limitation.
- **Future Directions**: We verify that each visual patch token encodes varying degrees of locality within the given image. Notably, visual patch tokens corresponding to the central part of the image tend to contain higher visual information compared to those at the edges, leading to an information discrepancy. Building on this observation, **selectively focusing on the more informative localized visual patch tokens could reduce the overall number of tokens, enhancing the efficiency of both training and inference.** Typically, increasing the number of visual tokens enhances visual perception capabilities. However, it also increases the final maximum sequence length of LLMs. For example, in the Qwen2-VL series, the number of visual patch tokens often exceeds 1,000, which can cause GPU Out-of-Memory (OOM) issues depending on the available computational resources. **By reducing the number of visual patch tokens based on the observed information discrepancy, we can mitigate such issues while maintaining performance.** For this, we can consider **adaptively reducing visual tokens based on the input image and text using Gumbel softmax, allowing the model to determine which visual tokens are important based on the given inputs, similar to DynamicViT [28].** Additionally, with fewer visual patch tokens, it becomes feasible to implement an interleaved format. This format is particularly practical for real-world applications, such as multi-modal conversations, where images can appear at arbitrary positions within the dialogue context. This future direction aligns with trends observed in prior studies, such as **Flamingo**, **HoneyBee**, and **BLIP-3**. While these studies have explored and adopted similar approaches, our work provides the first concrete evidence supporting this strategy. As such, our findings offer meaningful insights and lay a strong foundation for future research in this area.

---

> ### Author Response · Authors · 2024-11-24
> **Common Response for Reviewers (2/9)**
>
> ### **Findings 2: LLVMs are permutation invariant in terms of visual patch tokens.**
>
> A more detailed analysis across benchmarks reveals interesting patterns. In perception-related benchmarks, such as MMVP, Q-Bench, the performance drop is negligible. For LLaVA-1.5 and LLaVA-NeXT, performance even improves slightly in some cases. On the other hand, in text-rich benchmarks like MathVista and ChartQA, the performance drops significantly. **These benchmarks require understanding detailed numerical information and highly structured geometric graphs, where preserving the spatial structure of visual patch tokens is critical.**
>
> Interestingly, in the SQA-IMG benchmark, which includes science-related datasets, and the AI2D benchmark, which consists of diagram images, the relatively small performance gap is noteworthy, even though these images are rich in detail. We speculate that this phenomenon might be influenced by the difficulty of the benchmark, particularly the "question type." Benchmarks typically include two question formats: (1) free-form and (2) multiple-choice questions (MCQ). We hypothesize that:
>
> 1. LLMs can often solve questions using their extensive commonsense reasoning, even without image perception.
> 2. MCQ formats may be easier for models compared to free-form questions due to the presence of preferred answer patterns or inherent biases in selection.
>
> To investigate further, we conducted toy experiments comparing the difficulty of MathVista, ChartQA, SQA-IMG, and AI2D. We randomly selected 500 samples from each dataset and, for MCQ samples, included only those with four options. We then prompted ChatGPT (`gpt-3.5-turbo`) to answer these questions using the following templates:
>
> **MCQ:**
>
> ```
> Question: {question}
> Choices:
> {choices}
> E: I don't know.
>
> Please MUST generate only one option (A, B, C, D, E). Do not generate any explanation.
> Answer:
> ```
>
> **Free-Form:**
>
> ```
> Question: {question}
>
> Please provide your answer. If it is difficult to provide an answer, respond with "I don't know."
> ```
>
> We added the "I don't know" option to prevent the model from guessing randomly. The results are summarized below:
>
> | Dataset | Question Type | Accuracy (%) | "Don't Know" (%) |
> | --- | --- | --- | --- |
> | MathVista | Free-Form | 0.3 | 82.1 |
> |  | MCQ | 36.8 | 0 |
> |  | Overall | 13.6 | 52.2 |
> | ChartQA | Free-Form | 0 | 90 |
> | SQA-IMG | MCQ | 64.2 | 0 |
> | AI2D | MCQ | 53.2 | 1.6 |
>
> These results show that ChatGPT performs better on MCQ-type benchmarks compared to free-form types. Moreover, ChatGPT achieves higher accuracy on AI2D and SQA-IMG compared to MathVista and ChartQA. This supports the observation that LLVMs exhibit less permutation invariance in these text-rich benchmarks, possibly due to the nature of the datasets and their question formats. For free-form questions, the "don't know" response rate is significantly higher, indicating that these benchmarks are more challenging. **This highlights the need to minimize "blind" samples—questions solvable by LLMs without image perception—in benchmark design.** Benchmarks should prioritize free-form questions to reduce potential selection bias, as argued by recent studies like NaturalBench. (The above result is presented in Appendix E.1)
>
> - **Limitations:** As discussed, potential biases may arise due to the benchmarks' question types, making it risky to assert strong conclusions. This study primarily evaluates existing benchmarks, which feature both MCQ and free-form question types. Moreover, the scope of this work is limited to analyzing the behavior of LLaVA-family models, which, while popular, do not encompass the entire landscape of LLVMs. Expanding the evaluation to other models, such as Qwen2-VL (Appendix E.1), and incorporating additional benchmarks, like image captioning (Appendix E.7), would provide a more rigorous and comprehensive analysis.
> - **Future Directions:** Based on the findings, we suggest that future work focus on two key directions. First, it is essential to **develop more challenging benchmarks** that better explore LLVMs’ capabilities. Such benchmarks should prioritize free-form question types and avoid including "blind" samples that models can solve using commonsense reasoning without actually perceiving the image [1-2]. Building multi-turn interactive conversation benchmarks, like MMDU [3], could be particularly useful in this context. Second, since LLVMs generally exhibit permutation invariance, visual patch tokens can be treated as independent elements, allowing **images to be represented as unordered sets of points.** Applying paradigms like "Context Clusters," [4] which rely on clustering algorithms rather than convolutions or attention mechanisms, could improve interpretability and training efficiency. Furthermore, this approach could facilitate generalization to other data domains, such as point clouds [5], RGB-D data, or sensory images [6], broadening the applicability of LLVMs.

---

> ### Author Response · Authors · 2024-11-24
> **Common Response for Reviewers (3/9)**
>
> ### **Findings 3: LLVMs are sensitive to spatial structures.**
>
> In this finding, we randomly shuffled image patches to observe their impact on model performance. We found that LLaVA-OneVision is highly sensitive to spatial structures in text-rich benchmarks (e.g., MathVista, AI2D), which contain detailed numerical information. Conversely, it is relatively insensitive to spatial structures in perception-centric benchmarks (e.g., MMVP). We speculate that this result stems from the robustness property of vision encoders, as reported in a prior study [7], which indicate that ViT-variants are robust to spatial disruptions in datasets like ImageNet. However, unlike perception-related images, text-rich images include intricate numerical details. **Shuffling image patches in such cases disrupts the geometric relationships or the relative magnitudes in plots or charts, hindering the ability to interpret these images accurately.**
>
> Interestingly, LLaVA-1.5 and LLaVA-NeXT exhibit insensitivity to spatial structures, particularly on the MathVista dataset. We hypothesize that this might be due to the nature of the benchmark itself. Some instances in the dataset appear solvable even when the image is disrupted, likely because of question types that rely less on spatial accuracy. These results are discussed further in the preceding analysis and Section 3.4. Based on these observations, we argue that sensitivity to spatial structures varies as tasks shift from perception-related to reasoning-related.
>
> Specifically: For **perception-related tasks, LLVMs demonstrate robustness to spatial disruptions, likely benefiting from the inherent robustness of vision encoders**. For **reasoning-related tasks, LLVMs exhibit higher sensitivity to spatial structures, as these tasks require the integrity of spatial relationships for accurate interpretation.** This suggests a transition in dependency from the vision encoder to the LLM as tasks progress from perception to reasoning. For example, in perception tasks, LLVMs rely more on the vision encoder's capabilities. In contrast, in reasoning tasks, the LLM's reasoning power becomes more critical, as solving these complex tasks requires integrating perceived information from the given image.
>
> Based on these observations, we outline the following limitations and propose future directions:
>
> - **Limitations:** As discussed, a more comprehensive analysis would benefit from evaluating LLVMs using more challenging datasets that do not include instances solvable without image perception, such as OlympiadBench [30]. Additionally, expanding the evaluation to other LLVMs would strengthen our conclusions.
> - **Future Directions:** One future direction is to focus on building more challenging benchmarks. In such benchmarks, LLVMs should exhibit greater sensitivity to spatial structures because shuffled images would prevent them from solving complex questions effectively. Constructing these benchmarks would help better assess the models' true capabilities. Another direction is to develop more robust LLVMs that can handle spatial disruptions. Real-world images often lack perfect clarity—details may be missing, images may be flipped, or other disruptions may occur. To address this, we propose incorporating randomly shuffled images into the training process. By framing this as a **jigsaw puzzle task** [8], models can be trained to **reconstruct the original positions of image patches**. This approach could enhance their robustness to spatial variations, making them more applicable to real-world scenarios.

---

> ### Author Response · Authors · 2024-11-24
> **Common Response for Reviewers (4/9)**
>
> ### **Findings 4: In some cases, LLVMs can solve problems without seeing the image.**
>
> In this finding, we observe why LLaVA-1.5 and LLaVA-NeXT exhibit insensitivity to spatial structure (as noted in Findings 2). Specifically, even when presented with synthesized images that do not preserve the original numerical information, these models can solve math problems with minimal performance drop. In our experiment, we found that, in some instances within the MathVista dataset, LLVMs solve problems using the commonsense reasoning capabilities of their LLM backbone, such as answering questions like "*What is the smallest value?*" Based on these observations, we outline the following limitations and propose future directions:
>
> - **Limitations:** The synthesized images were generated using LLaVA-OneVision-7B with the prompt template "*Please generate a caption of this image*" and SDXL-Lightning. To obtain more robust results, future experiments should explore captions with varying levels of detail—ranging from concise to highly detailed—by using alternative prompt templates, more specialized captioning models (e.g., ShareCaptioner [29]), or more advanced text-to-image generation models that outperform SDXL-Lightning. Incorporating these variations would strengthen the reliability of our conclusions.
> - **Future Directions:** To assess whether LLVMs genuinely solve math or chart-related problems by interpreting visual content or leveraging real numerical information from the images, it is necessary to develop more challenging benchmarks. **These benchmarks should ensure that LLMs (without perceiving the image) cannot solve visual math problems, thereby isolating and testing the visual reasoning capabilities of LLVMs more effectively.**
>
> ---
>
> ### **Findings 5: Scaling up visual instruction tuning datasets improves text-only mathematical reasoning.**
>
> In this finding, we observe that scaling up visual instruction tuning datasets enhances both visual and text-based mathematical reasoning. Based on these observations, we outline the following limitations and propose future directions:
>
> - **Limitations:** In this experiment, we used 8-shot Chain-of-Thought (CoT) prompting on the GSM8K dataset. While we observed a scaling effect, the observed correlation might be spurious. This is because visual instruction tuning datasets often contain diverse information types, such as charts, code, documents, and OCR-related data. The improved performance may not solely result from the increased size of the visual instruction tuning dataset but could also depend on various factors, including the training recipe (e.g., number of training epochs, learning rate, optimizer), architectural design choices, or the number of visual tokens. Therefore, we refrain from making a strong argument that scaling up visual instruction tuning datasets is the most influential factor in the compatibility between text-only and visual mathematical reasoning. Instead, we suggest that latent factors may contribute to this effect, with scaling up the visual instruction tuning dataset being one of them. To strengthen our observations, future experiments should include additional text-only math datasets with different prompt settings, such as varying the number of few-shot examples and the diversity of demonstrations. As reported in prior studies [31,32], prompt choice can significantly influence performance (i.e., prompt sensitivity). Additionally, testing other LLVMs, such as Qwen2-VL, would provide further insights.
> - **Future Directions:** Our findings suggest that increasing the size of visual instruction tuning datasets containing math, diagram, and chart-related images with specific rationales could enhance text-only mathematical reasoning. This aligns with recent data-centric AI perspectives [27], which emphasize that **improving the quality and diversity of datasets (even on a small scale) can lead to better-aligned and higher-performing models.** Based on these results, researchers developing new LLVMs should prioritize preparing high-quality (e.g., rationale-driven) and diverse (e.g., chart, math, code) visual instruction tuning datasets. Such datasets would likely enhance the cognitive reasoning and perceptual capabilities of the trained models.

---

> ### Author Response · Authors · 2024-11-24
> **Common Response for Reviewers (5/9)**
>
> ### **Findings 6: LLVMs are robust against occlusion.**
>
> In this experiment, we sought to examine whether the well-known robustness of ViT-variant encoders under occlusion, compared to their CNN counterparts, also applies to LLVMs—specifically, whether the robustness properties of ViT-based vision encoders transfer to LLVMs. We observed that LLVMs exhibit relatively strong performance under occlusion. For instance, in the AI2D dataset, even when 50-70% of patches are missing, LLVMs can still provide correct answers to some extent. Based on these results, we outline the following limitations and propose future directions:
>
> - **Limitations:** As mentioned above, the ability of LLVMs to solve problems in occluded settings might be influenced by the question type in the benchmark dataset (e.g., AI2D is a MCQ dataset). To conduct a more rigorous analysis, future experiments should test on other visual processing datasets, such as image captioning datasets. Additionally, this experiment was conducted exclusively on the LLaVA family of models; evaluating other LLVMs would help solidify our observations. Furthermore, during patch-dropping, we used the `dino-small` model for both `Salient PatchDrop` and `Non-Salient PatchDrop`. The impact of patch dropping may vary depending on the size and type of self-supervised vision model used, potentially leading to different patterns of performance degradation.
> - **Future Directions:** Building more challenging benchmarks would be an important direction for future work. Moreover, real-world images on the web often contain noise or missing details (e.g., partially obscured or incomplete images). Training LLVMs to handle such scenarios by incorporating contextual reasoning or abductive reasoning could be highly beneficial. For instance, in a chart problem where the fourth bar in a bar chart is missing, humans can infer its approximate size based on the information from the other three bars. Similarly, training LLVMs to infer missing details in such cases could enhance their real-world applicability and robustness.

---

> ### Author Response · Authors · 2024-11-24
> **Common Response for Reviewers (6/9)**
>
> ### **Findings 7: LLVMs struggle to preserve the original visual understanding capability.**
>
> In this finding, we observe that after visual instruction tuning, the visual perception capability of the original vision encoder in LLVMs may degrade. We hypothesize that training LLVMs to solve complex tasks (e.g., chart or math reasoning) using instructions could lead to a loss of basic perception abilities (e.g., recognizing simple objects), a phenomenon known as "**catastrophic forgetting.**" Additionally, due to discrepancies in parameter sizes between the vision encoder and the LLM, training a simple linear layer (projector) to bridge the gap may result in an imbalance where the LLM dominates. Based on these observations, we outline the following limitations and propose future directions:
>
> - **Limitations:** While we evaluated visual perception capabilities across various image datasets, there are many domain-specific image datasets in the real world. To derive generalizable conclusions, evaluating on additional datasets, such as the VTAB benchmark [33], would be beneficial. In this experiment, we investigated catastrophic forgetting by following existing experimental setups [34]. However, comparing LLVMs with contrastive approaches may seem unfair due to multiple factors influencing LLVM performance, such as prompt variations and the method used to calculate accuracy from generated text. For a more rigorous analysis, it is necessary to explore different prompt methods and finetune LLVMs on zero-shot image classification datasets (e.g., CIFAR-100), then evaluate whether perception capability improves. Regarding the LLM-dominance problem during visual instruction tuning, confirming this phenomenon is challenging. To test it effectively, one would need to train LLVMs with the same LLM size but varying vision encoder scales using identical datasets. Alternatively, we could evaluate other types of LLVMs that incorporate external computer-vision models (e.g., segmentation models), such as MoAI [35]. Using visually enhanced LLVMs would strengthen our argument.
> - **Future Directions:** Balancing perception and cognitive reasoning capabilities is critical. The "catastrophic forgetting" problem [9] has been a long-standing issue in machine learning. A standard approach is to train models on mixed datasets [10-11] with a carefully designed balance (a "golden ratio") between perception- and reasoning-related data. **Continuously training LLVMs** on perception-focused datasets following **rehearsal methods** [12] can minimize catastrophic forgetting by retaining knowledge of prior tasks while learning new ones. **Knowledge distillation [13] from large-scale LLVMs (e.g., 72B parameters) to smaller-scale models (e.g., 7B parameters) could help preserve perception capabilities while maintaining reasoning strength**. Alternatively, fine-tuning **adapters** (e.g., p-tuning [14], LoRA [15], Q-LoRA [16]) on task-specific datasets offers a lightweight solution to improve performance on new tasks without sacrificing existing capabilities.

---

> ### Author Response · Authors · 2024-11-24
> **Common Response for Reviewers (7/9)**
>
> ### **Findings 8: LLVMs lose the ability to understand and interpret shared world concepts.**
>
> In this finding, we observe that cross-modal alignment is diminished after visual instruction tuning, supporting the platonic representation hypothesis. To investigate this, we utilized the DOCCI dataset, which includes natural and realistic images accompanied by long, dense captions containing nine key visual features (e.g., attributes, text rendering, views, scenes, optical effects). Using this dataset, we assessed cross-modal alignment in terms of localized visual understanding. Based on these observations, we outline the following limitations and propose future directions:
>
> - **Limitations:** To evaluate the generalizability of these findings, it would be beneficial to test cross-modal alignment on a wider variety of datasets, such as CC12M, WIT, and RedCaps12M. Additionally, evaluating this phenomenon on various LLVMs, such as LLaVA-OneVision and Qwen2-VL, would provide a more comprehensive understanding.
> - **Future Directions:** Maintaining the original cross-modal alignment is critically important. Continual learning methods could be applied to mitigate the loss of alignment during visual instruction tuning. **Enhancing the visual perception capability of the projector during training could also help.** For instance, employing models like **HoneyBee** [17], which incorporate convolution layer-based projectors, could improve localized understanding. Convolution layers are well-known for their strong inductive bias toward localized feature extraction, making them better suited for capturing fine-grained details in images. Even with the inclusion of complex instruction datasets (e.g., charts, math), a carefully designed projector that excels at extracting detailed and localized information from images could naturally improve both perception and reasoning capabilities. We hypothesize that **enhancing localized perception would inherently lead to improvements in reasoning, aligning the two capabilities more effectively.**
>
> ---
>
> ### **Findings 9: LLVMs strongly focus on the center of the image.**
>
> In this finding, we observe that not all visual tokens are equally important, as determined by our defined metric of "importance score." Specifically, visual tokens corresponding to the central part of the image are more actively utilized for both perception and reasoning tasks. Based on these observations, we outline the following limitations and propose future directions:
>
> - **Limitations:** For a comprehensive analysis, measuring the importance score for all visual tokens (e.g., 576 tokens for LLaVA-1.5) across various datasets would be ideal. However, as mentioned in our paper, conducting 576 evaluations for a single run is computationally expensive, and repeating the process K-times to ensure robustness would require significant resources. To address this challenge, we adopted a group-wise token analysis on LLaVA-1.5 using the MM-Vet dataset only. MM-Vet was chosen because it is an integrated benchmark that includes samples requiring both perception and reasoning capabilities. However, for a more rigorous analysis, evaluating other LLVMs on diverse benchmarks using all visual tokens would be highly beneficial.
> - **Future Directions:** Based on our observations, reducing redundant visual tokens in the projector could enhance training and inference efficiency, aligning with findings from prior studies [17-18]. Typically, the large number of visual tokens poses a computational burden. This is particularly relevant for real-world scenarios where interleaved format-style conversations [20] are predominant. High visual token counts can make it challenging to train more effective LLVMs for such interleaved conversational formats. Our findings provide a practical direction for reducing visual token counts while maintaining performance. By doing so, we can enable the training of interleaved-format LLVM models more efficiently, similar to approaches highlighted in previous researches [18-19].

---

> ### Author Response · Authors · 2024-11-24
> **Common Response for Reviewers (8/9)**
>
> ### **Findings 10: Lower block layers in LLVMs are more important.**
>
> In this finding, we observe that the lower layers (< 6) of LLVMs play a crucial role in handling the integrated capabilities required for tasks in the MMStar dataset. Based on this observation, we outline the following limitations and propose future directions:
>
> - **Limitations:** For a more rigorous analysis, it would be beneficial to conduct this experiment across various LLVMs using a wider range of evaluation benchmarks. Additionally, we could investigate whether this phenomenon is specific to LLVMs or if it is inherent to the general architecture of single-modality LLMs.
> - **Future Directions:** Based on our observations, we emphasize the importance of the traversing layers (TroL) approach, as proposed in the TroL paper [21], in improving generalization. In this approach, models are trained to revisit and leverage layer-specific information during the training process. The paper demonstrates that **lower layers are more actively engaged, which aligns with our findings.** These results suggest that the lower layers of LLVMs play a critical role in establishing a foundational understanding of the world. To enhance this capability, increasing the signal for world understanding in the lower layers during training could be a promising direction. One potential method is **injecting noise information into the lower layers during training**, as suggested in a prior study [22]. This technique could improve the robustness of LLVMs, further solidifying their foundational perception and reasoning capabilities.
>
> ---
>
> ### **Findings 11: Textual modality is more important than visual modality.**
>
> In this finding, we observe that LLVMs initially focus on global image perception, but by the middle layers, they have processed the image and shifted their attention toward solving complex user queries to generate coherent answers. Based on these observations, we outline the following limitations and propose future directions:
>
> - **Limitations:** For a more rigorous analysis, it would be beneficial to conduct this experiment across various LLVMs and a wider range of evaluation benchmarks to validate the findings comprehensively. Similar to Findings 10, we need more detailed investigation to determine whether this observation stems from the architectural design of specific LLVMs or emerges from domain-specific training objectives. For example, in text-to-image generation tasks, the image modality might play a more dominant role.
> - **Future Directions:** While the textual modality appears more influential in higher layers, improving the visual perception capability in lower layers is crucial. This is because LLVMs rely heavily on understanding the given image during the initial processing stages. As suggested in prior works [17,23], using a larger number of visual tokens, adopting high-resolution image processing [24], or employing dynamic image processing methods [25] is essential for enhancing performance. Furthermore, **strengthening the projector's capability for localized visual understanding could be beneficial.** For instance, after the initial image-caption alignment step (commonly the first step in LLVM training), an additional training phase called **"empowering localized understanding"** could be introduced before visual instruction tuning. This phase would involve adding an extra layer, referred to as the "AL" (Augmented Layer), on top of the simple linear layer. The **AL would be trained using a masked autoencoder (MAE) approach [26], where the model learns to predict masked image patches.** This process would enhance localized visual understanding, ultimately improving the balance between visual and textual modalities and boosting overall model performance.
>
> ---
>
> **References**
>
> [1] Fu, Xingyu, et al. "Blink: Multimodal large language models can see but not perceive." *European Conference on Computer Vision*. Springer, Cham, 2025.
>
> [2] Li, Baiqi, et al. "NaturalBench: Evaluating Vision-Language Models on Natural Adversarial Samples." *arXiv preprint arXiv:2410.14669* (2024).
>
> [3] Liu, Ziyu, et al. "MMDU: A Multi-Turn Multi-Image Dialog Understanding Benchmark and Instruction-Tuning Dataset for LVLMs." *arXiv preprint arXiv:2406.11833* (2024).

---

> ### Author Response · Authors · 2024-11-24
> **Common Response for Reviewers (9/9)**
>
> [4] Ma, Xu, et al. "Image as set of points." *arXiv preprint arXiv:2303.01494* (2023).
>
> [5] Ma, Xu, et al. "Rethinking network design and local geometry in point cloud: A simple residual MLP framework." *arXiv preprint arXiv:2202.07123* (2022).
>
> [6] Yu, Youngjoon, et al. "Spark: Multi-vision sensor perception and reasoning benchmark for large-scale vision-language models." *arXiv preprint arXiv:2408.12114* (2024).
>
> [7] Naseer, Muhammad Muzammal, et al. "Intriguing properties of vision transformers." *Advances in Neural Information Processing Systems* 34 (2021): 23296-23308.
>
> [8] Chen, Yingyi, et al. "Jigsaw-vit: Learning jigsaw puzzles in vision transformer." *Pattern Recognition Letters* 166 (2023): 53-60.
>
> [9] Kirkpatrick, James, et al. "Overcoming catastrophic forgetting in neural networks." *Proceedings of the national academy of sciences* 114.13 (2017): 3521-3526.
>
> [10] Ke, Zixuan, Bing Liu, and Xingchang Huang. "Continual learning of a mixed sequence of similar and dissimilar tasks." *Advances in neural information processing systems* 33 (2020): 18493-18504.
>
> [11] Gururangan, Suchin, et al. "Don't stop pretraining: Adapt language models to domains and tasks." *arXiv preprint arXiv:2004.10964* (2020).
>
> [12] Rebuffi, Sylvestre-Alvise, et al. "icarl: Incremental classifier and representation learning." *Proceedings of the IEEE conference on Computer Vision and Pattern Recognition*. 2017.
>
> [13] Jin, Xisen, et al. "Lifelong pretraining: Continually adapting language models to emerging corpora." *arXiv preprint arXiv:2110.08534* (2021).
>
> [14] Liu, Xiao, et al. "P-tuning v2: Prompt tuning can be comparable to fine-tuning universally across scales and tasks." *arXiv preprint arXiv:2110.07602* (2021).
>
> [15] Hu, Edward J., et al. "Lora: Low-rank adaptation of large language models." *arXiv preprint arXiv:2106.09685* (2021).
>
> [16] Dettmers, Tim, et al. "Qlora: Efficient finetuning of quantized llms." *Advances in Neural Information Processing Systems* 36 (2024).
>
> [17] Cha, Junbum, et al. "Honeybee: Locality-enhanced projector for multimodal llm." *Proceedings of the IEEE/CVF Conference on Computer Vision and Pattern Recognition*. 2024.
>
> [18] Xue, Le, et al. "xgen-mm (blip-3): A family of open large multimodal models." *arXiv preprint arXiv:2408.08872* (2024).
>
> [19] Li, Feng, et al. "Llava-next-interleave: Tackling multi-image, video, and 3d in large multimodal models." *arXiv preprint arXiv:2407.07895* (2024).
>
> [20] Lee, Young-Jun, et al. "Stark: Social Long-Term Multi-Modal Conversation with Persona Commonsense Knowledge." *arXiv preprint arXiv:2407.03958* (2024).
>
> [21] Lee, Byung-Kwan, et al. "TroL: Traversal of Layers for Large Language and Vision Models." *arXiv preprint arXiv:2406.12246* (2024).
>
> [22] Jain, Neel, et al. "Neftune: Noisy embeddings improve instruction finetuning." arXiv preprint arXiv:2310.05914 (2023).
>
> [23] McKinzie, Brandon, et al. "MM1: methods, analysis and insights from multimodal LLM pre-training." *European Conference on Computer Vision*. Springer, Cham, 2025.
>
> [24] Liu, Haotian, et al. "Improved baselines with visual instruction tuning." *Proceedings of the IEEE/CVF Conference on Computer Vision and Pattern Recognition*. 2024.
>
> [25] Wang, Peng, et al. "Qwen2-vl: Enhancing vision-language model's perception of the world at any resolution." *arXiv preprint arXiv:2409.12191* (2024).
>
> [26] He, Kaiming, et al. "Masked autoencoders are scalable vision learners." *Proceedings of the IEEE/CVF conference on computer vision and pattern recognition*. 2022.
>
> [27] Zhou, Chunting, et al. "Lima: Less is more for alignment." Advances in Neural Information Processing Systems 36 (2024).
>
> [28] Rao, Yongming, et al. "Dynamicvit: Efficient vision transformers with dynamic token sparsification." *Advances in neural information processing systems* 34 (2021): 13937-13949.
>
> [29] Chen, Lin, et al. "Sharegpt4v: Improving large multi-modal models with better captions." *arXiv preprint arXiv:2311.12793* (2023).
>
> [30] He, Chaoqun, et al. "Olympiadbench: A challenging benchmark for promoting agi with olympiad-level bilingual multimodal scientific problems." *arXiv preprint arXiv:2402.14008* (2024).
>
> [31] Ye, Seonghyeon, et al. "Guess the instruction! flipped learning makes language models stronger zero-shot learners." *arXiv preprint arXiv:2210.02969* (2022).
>
> [32] Lu, Yao, et al. "Fantastically ordered prompts and where to find them: Overcoming few-shot prompt order sensitivity." *arXiv preprint arXiv:2104.08786* (2021).
>
> [33] Zhai, Xiaohua, et al. "A large-scale study of representation learning with the visual task adaptation benchmark." *arXiv preprint arXiv:1910.04867* (2019).
>
> [34] Zhai, Yuexiang, et al. "Investigating the catastrophic forgetting in multimodal large language models." *arXiv preprint arXiv:2309.10313* (2023).
>
> [35] Lee, Byung-Kwan, et al. "Moai: Mixture of all intelligence for large language and vision models." *European Conference on Computer Vision*. Springer, Cham, 2025.

---

### Author Response · Authors · 2024-11-24
**General Response for Reviewers**

We sincerely thank all reviewers for their valuable feedback, which has been instrumental in improving the quality of our paper. Specifically, the comments on clarifying our findings and discussing the generalizability of our results from both model and dataset perspectives have been particularly helpful. Based on the reviewers' feedback, we have made the following updates to the paper:

### Main Paper

- **Additional Interpretations**: We have added further explanations and interpretations, highlighted in blue for clarity.
    - We added more detailed results in Section 3.3 regarding benchmark-dependent "permutation invariance," providing a more precise analysis.
    - We added an explanation for why LLVMs are sensitive to spatial structures in text-rich benchmarks (e.g., MathVista, AI2D) in Section 3.3.
    - We clarify in Table 2 whether "Freq." belongs to "Orig." or "Syn."
    - We conducted five experimental runs of the zero-shot image classification task and updated the results in Table 3.
- **Adjustments for Page Limit**: To comply with the 10-page limit, we resized Figures and Tables where necessary and moved the related work regarding "model-stitching" to Appendix F.
    - We reduced the size of Figure 4, Table 3, and Figure 7.

### Appendix

- **Limitations and Further Discussions**: In Appendix A, we added the limitations of our experiments and included further discussions for each of our findings.
- **Platonic Representation Hypothesis**: Appendix B now includes a detailed explanation of the "platonic representation hypothesis," how cross-modal alignment is measured using our alignment metric, and the motivation for selecting the DOCCI dataset.
- **Importance Score**: In Appendix C, we provide a comprehensive explanation of the "importance score," including its calculation methodology.
- **Experimental Setup**: Appendix D includes detailed descriptions of our experimental setup and specific conditions necessary to derive our findings.
- **Generalization of Findings**: In Appendix E, we present additional experimental results to strengthen the generalizability of our findings. These include evaluations of Qwen2-VL and Fuyu-8B models, experiments using image-captioning datasets, and detailed analyses of our observations.
- **More Examples:** In Appendices G, H, and I, we present additional examples of synthesized, shuffled, and occluded images used in our experiments.

We deeply appreciate the reviewers' insightful suggestions. While we acknowledge that extending our observations to additional LLVMs would further enhance the rigor of our study, we focused on models such as the LLaVA family and Qwen2-VL, which are currently recognized as significant LLVMs. We believe that our observations on these models are meaningful and impactful. Moving forward, we are committed to expanding our research, and—if we secure sufficient resources—plan to include additional experiments on specific benchmarks that require access to OpenAI API or other large-scale resources.

Once again, we thank all reviewers for their constructive feedback and for helping us refine our work.

---

### Author Response · Authors · 2024-12-01
**Friendly Reminder**

Dear Reviewers, this is a friendly reminder that the reviewer discussion period concludes on December 2. We would be sincerely grateful to receive your feedback at your earliest convenience. Your insights are invaluable to us. If you need any additional information or have further questions, please do not hesitate to reach out. Thank you very much for your time and thoughtful consideration!

---

### Meta-Review · Area_Chair_R9qs · 2024-12-22

**Metareview:**

The paper considers a critical problem of how LLVMs perceive and process visual information and aims to provide a comprehensive study relevant to the community through a diverse set of benchmarks and experiments. In principle, findings could inform the future models and benchmarks. While the paper provides valuable insights, the reviewers find that conclusions were overgeneralized, lacked methodological justification, and the implications were limited to a limited set of models. While the rebuttal phase brought improvements and strengthened the work, the AC feels that a major revision is required to incorporate all these findings.

**Additional Comments On Reviewer Discussion:**

Excellent discussion with the authors, but given the amount of changes required, I will suggest the authors to incorporate these and submit the major revision to a follow-up conference.

---

### Decision · Program_Chairs · 2025-01-22

Reject